# CONSIS-GCPO: CONSISTENCY-PRESERVING GROUP CAUSAL PREFERENCE OPTIMIZATION FOR VISION CUSTOMIZATION

**Qiaoqiao Jin**[1][*] **Dong She**[2][*], **Siming Fu**[3][*][†], **Hualiang Wang**[4], **Jidong Jiang**[5]
[1]Shanghai Jiao Tong University   [2]University of Science and Technology of China
[3]Zhejiang University   [4]The Hong Kong University of Science and Technology [5] Nanjing University
{jinqiaoqiao}@sjtu.edu.cn

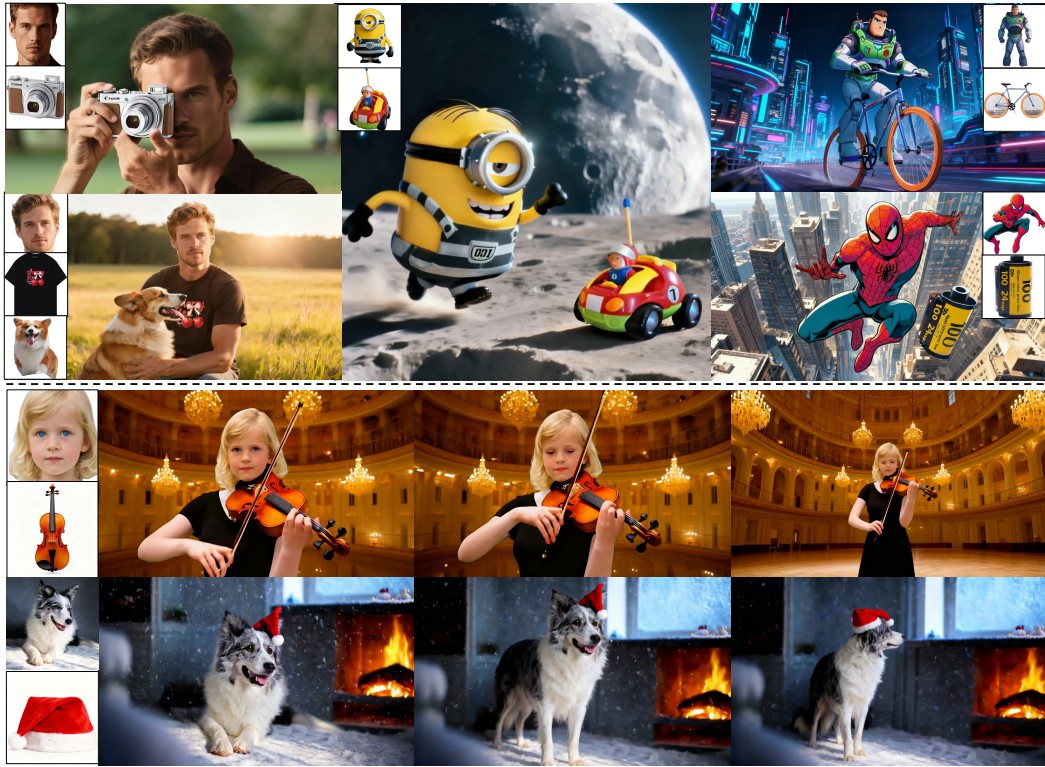

Figure 1: **Consis-GCPO** achieves high-quality personalized generation across diverse scenarios. **Top:** R2I generation with complex multi-subject compositions and interactions. **Bottom:** R2V generation demonstrating temporal consistency and subject fidelity in motion sequences. Corresponding prompts and additional visualization results are provided in the **Appendix**.

## ABSTRACT

Subject-driven generation faces a fundamental challenge: achieving high subject fidelity while maintaining semantic alignment with textual descriptions. While recent GRPO-based approaches have shown promise in aligning generative models with human preferences, they apply uniform optimization across all denoising timesteps, ignoring the temporal dynamics of how textual and visual conditions influence generation. We present **Consis-GCPO**, a causal reinforcement learning framework that reformulates multi-modal condition generation through discrete-time causal modeling. Our **key insight** is that different conditioning signals exert varying influence throughout the denoising process—text guides semantic structure in early steps while visual references anchor details in later stages. By introducing decoupled causal intervention trajectories, we quantify instantaneous

---

[*]Equal contribution.   [†]Corresponding author.

causal effects at each timestep, transforming these measurements into temporally-weighted advantages for targeted optimization. This approach enables precise tracking of textual and visual contributions, ensuring accurate credit assignment for each conditioning modality. Extensive experiments demonstrate that Consis-GCPO significantly advances personalized generation, achieving superior subject consistency while preserving strong text-following capabilities, particularly excelling in complex multi-subject scenarios.

# 1 INTRODUCTION

Personalized content creation (Ruiz et al., 2023; Wu et al., 2025; Mou et al., 2025; Chen et al., 2025; She et al., 2025; Jiang et al., 2024) has emerged as a critical capability in generative modeling, enabling users to synthesize diverse outputs that maintain consistency with provided references. In the image domain, reference-to-image (R2I) generation has achieved remarkable progress through various adaptation strategies, from full-model fine-tuning and parameter-efficient techniques like LoRA (Hu et al.) to lightweight conditioning mechanisms using subject embeddings. Recent R2I methods such as DreamO (Mou et al., 2025), XVerse (Chen et al., 2025), and MOSAIC She et al. (2025) have further advanced the field by introducing multi-reference subject generation, enabling consistent synthesis across multiple input references. Building upon these advances, reference-to-video (R2V) generation extends personalization to the temporal domain, with frameworks like VACE Jiang et al. (2025) and Phantom Liu et al. (2025c) demonstrating compelling results for both single- and multi-subject video generation while preserving identity consistency across frames.

Despite these technical achievements, current approaches face fundamental limitations in balancing competing objectives that prevent them from meeting human preferences as shown in Figure 2. **Subject fidelity degradation** manifests when models struggle to preserve fine-grained characteristics and identity consistency, particularly in complex multi-subject compositions where interactions between subjects must be maintained. Concurrently, **semantic alignment drift** emerges as visual reference conditioning often interferes with the model's text-following capabilities, causing generated content to prioritize visual similarity at the expense of semantic coherence. These dual challenges result in a problematic trade-off: models either produce visually accurate outputs that ignore textual instructions or semantically correct generations that fail to maintain subject identity. This inability to simultaneously satisfy both visual consistency and semantic alignment creates significant barriers to real-world deployment.

Recent advances in reinforcement learning for generation, particularly Group Relative Policy Optimization (GRPO) methodologies like Flow-GRPO (Liu et al., 2025a) and DanceGRPO (Xue et al., 2025), have demonstrated success in aligning models with human preferences, offering a promising avenue to address these challenges. **However**, these approaches suffer from critical shortcomings when applied to multimodal generation: **temporal blindness**—they ignore how the importance of textual versus visual conditioning varies across denoising timesteps, applying uniform optimization throughout the generation trajectory; and **entangled feedback**—they provide only terminal rewards without decomposing the individual contributions of text and reference conditions, making targeted improvements impossible.

To address these fundamental limitations, we propose **Consis-GCPO**, a principled framework that reformulates multi-condition guided generation as a temporal causal optimization problem. The foundation of Consis-GCPO is a discrete-time structural causal model (SCM) that explicitly models causal dependencies between conditioning signals and generation outcomes at each denoising timestep—addressing the critical gap in existing methods' inability to capture temporal dynamics of multimodal conditions. Leveraging this causal foundation, Consis-GCPO implements decoupled causal interventions through targeted ablations that selectively remove specific conditions at individual timesteps. By comparing main generation trajectories with prompt and reference-intervention paths, we precisely quantify when textual semantics versus visual references are most critical during denoising. These instantaneous causal effects are converted into temporally-weighted advantages that enable dynamic adjustment of semantic alignment and subject fidelity based on each timestep's actual importance rather than uniform treatment. The resulting optimization adaptively integrates textual and visual guidance based on measured temporal contributions rather than heuristic weighting schemes. **Notably**, our method achieves superior multimodal consistency while maintaining

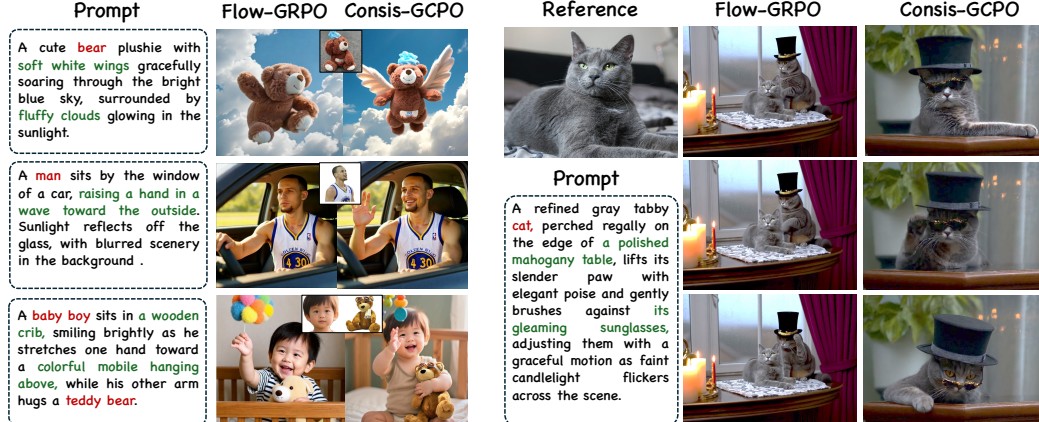

Figure 2: **Challenges in subject-driven generation.** Current approaches struggle to simultaneously achieve subject fidelity and semantic alignment in R2I (left) and R2V (right). Our Consis-GCPO addresses these limitations through temporally-aware causal interventions that optimize conditioning effects across different denoising timesteps.

computational efficiency, particularly excelling in complex multi-subject scenarios that current approaches struggle with. Our contributions are as follows:

- We identify the fundamental limitation of existing GRPO methods in subject-driven generation—the inability to capture temporal dynamics of conditioning signals. Our proposed **Consis-GCPO** addresses this through causal reformulation and temporal intervention to estimate causal effect of visual and textual conditions, enabling precise quantification of when textual semantics versus visual references are most critical during denoising.

- We introduce **decoupled causal intervention** and **temporally-weighted advantage computation** mechanisms that transform instantaneous causal effects into targeted optimization signals. This enables the model to dynamically adjust semantic alignment and subject fidelity based on each timestep's actual importance rather than uniform treatment.

- We demonstrate significant improvements over state-of-the-art personalized generation methods, achieving superior subject consistency while preserving text-following capabilities. Our approach particularly excels in complex scenarios involving multiple subjects and intricate interactions, validating the effectiveness of causal-guided optimization.

## 2 RELATED WORKS

**Subject-Driven Generation**    Subject-driven generation aims to synthesize visual content that preserves reference subject identity while following textual descriptions. Early works like Dream-Booth (Ruiz et al., 2023), IP-Adapter (Ye et al., 2023) and MS-Diffusion (Wang et al., 2025) achieved subject consistency through fine-tuning or attention mechanisms in UNet architectures, while recent transformer-based approaches including UNO (Wu et al., 2025) and XVerse (Chen et al., 2025) leverage in-context learning for enhanced subject preservation. Video generation introduces additional temporal consistency challenges. MAGREF (Deng et al., 2025) addresses multi-subject video synthesis through masked guidance mechanisms, Phantom proposes unified text-image injection for cross-modal alignment, and VACE (Jiang et al., 2025) introduces an all-in-one framework unifying generation and editing tasks. Despite these advances, existing methods struggle with balancing textual adherence and visual consistency, particularly in multi-subject scenarios where maintaining individual subject fidelity across temporal sequences remains challenging.

**Reinforcement learning for image generation**    Reinforcement learning has recently emerged as an effective paradigm for enhancing text-to-image generation models. Flow-GRPO (Liu et al., 2025a) first integrated online RL into flow matching models through ODE-to-SDE conversion,

demonstrating substantial improvements in compositional generation and visual text rendering tasks. DanceGRPO (Xue et al., 2025) adapts GRPO for visual generation, achieving consistent and stable policy optimization across diffusion models and rectified flows while scaling effectively to large and diverse prompt sets. These works collectively demonstrate that RL-based optimization effectively addresses fundamental challenges in text-to-image generation, particularly in improving text adherence, compositional understanding, and human preference alignment while maintaining computational efficiency and preventing reward hacking behaviors.

## 3 PRELIMINARY: FLOW-GRPO

The starting point of our approach is the training of a conditional flow model for reference-to-vision generation (image or video). The typical input unit contains three components: (1) the textual prompt $P$, which specifies the human instruction; (2) the reference images $I_r$, which inject the intended visual identity or style; and (3) the noised latent representation, defined as:

$$\boldsymbol{x}_t = (1 - t)\boldsymbol{x}_0 + t\boldsymbol{x}_1; \quad t \in [0, 1], \tag{1}$$

where $\boldsymbol{x}_0$ denotes the target image and $\boldsymbol{x}_1$ is a random noise sample. The conditional flow model $\boldsymbol{v}_\theta(\cdot)$ is then defined and trained to predict the velocity field that transports $\boldsymbol{x}_t$ towards $\boldsymbol{x}_0$.

**SDE-based iterative denoising** Once trained, the conditional flow model can be applied in the generative stage by iteratively denoising a random initialization. To enable sampling diversity and facilitate richer exploration during generation, we adopt a Stochastic Differential Equation (SDE) formulation:

$$\boldsymbol{x}_{t-\Delta t} = f_\theta(\boldsymbol{x}_t, P, I_r, \epsilon_t) = \boldsymbol{x}_t - \Delta t \cdot \boldsymbol{v}_\theta(\boldsymbol{x}_t, t, P, I_r) + g(t)\sqrt{\Delta t}\,\epsilon_t, \tag{2}$$

where $\epsilon_t \sim \mathcal{N}(0, \mathbf{I})$ is an i.i.d. Gaussian random variable introduced at each denoising step, and $g(t)$ is a time-dependent diffusion coefficient. Equivalently, the transition distribution can be written as

$$\boldsymbol{x}_{t-\Delta t} \sim \mathcal{N}\Big(\boldsymbol{x}_t - \Delta t \cdot \boldsymbol{v}_\theta(\boldsymbol{x}_t, t, P, I_r),\, g^2(t)\Delta t \cdot \mathbf{I}\Big). \tag{3}$$

This stochastic formulation augments the deterministic ODE trajectory with Gaussian perturbations, thereby encouraging sampling diversity while still preserving conditional guidance.

On top of SDE-based denoising, Flow-GRPO (Liu et al., 2025a) enhances flow-based generative models through online reinforcement learning. The method treats the denoising process as a sequential decision problem, where the policy is defined as $\pi(t) \triangleq p_\theta(\boldsymbol{x}_{t-\Delta t}|\boldsymbol{x}_t)$ in Eqn 3. Formally, the Flow-GRPO loss is defined as:

$$\mathcal{L}_\theta = \frac{1}{G}\sum_{g=1}^{G}\Delta t\sum_{t=1}^{t=\Delta t}(min(r_t^g(\theta)\mathcal{A}_t^g, \mathrm{clip}(r_t^g(\theta), 1-\sigma, 1+\sigma)\mathcal{A}_t^g) - \beta D_{KL}(\pi_\theta||\pi_{\mathrm{ref}})), \tag{4}$$

where $r_t^g(\theta) = \frac{p_\theta(\boldsymbol{x}_{t-\Delta t}^g|\boldsymbol{x}_t^g)}{p_{\theta_{\mathrm{old}}}(\boldsymbol{x}_{t-\Delta t}^g|\boldsymbol{x}_t^g)}$ and $\Delta t$ is the interval of inter time steps and clip is a clamp function to restrict the value to the range $[1-\sigma, 1+\sigma]$.

## 4 DISCRETE-TIME CAUSAL MODELING FOR MULTI-CONDITION GUIDED GENERATION

We reformulate the vision customization generation task through the lens of causal inference, enabling principled analysis of how textual and visual conditioning jointly influence the generation process. Specifically, at each discrete timestep $t$ in the reverse diffusion process, we model the denoised state $\boldsymbol{x}_{t-\Delta t}$ as causally determined by four parent variables: the current noisy latent $\boldsymbol{x}_t$, the textual prompt $P$, the reference image $I_r$, and an independent noise term $\epsilon_t$. This relationship defines our structural causal model (SCM) for single-step denoising, i.e., $(\boldsymbol{x}_t, P, I_r) \to \boldsymbol{x}_{t-\Delta t}$.

To quantify individual modal contributions, we employ causal interventions through targeted ablations. Unlike global ablation methods, we introduce *step-wise causal interventions* that isolate causal effects at specific timesteps.

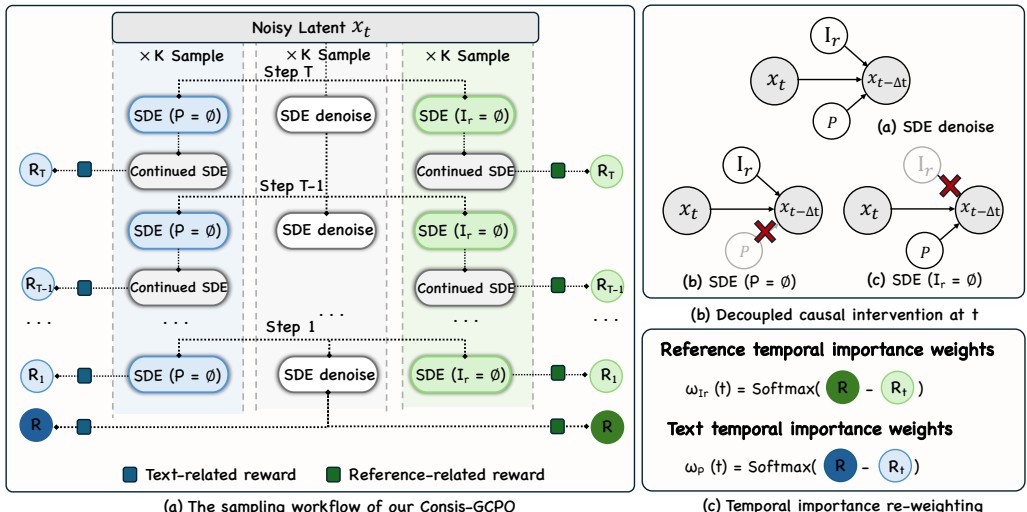

Figure 3: **Overview of Consis-GCPO framework.** (a) Sampling workflow showing step-wise counterfactual interventions where prompt $P$ or reference $I_r$ are selectively ablated at specific timesteps during SDE denoising, generating multiple trajectories for causal effect quantification. (b) Decoupled causal intervention at timestep t, illustrating how ablating prompt $P$ or reference $I_r$ enables isolation of individual conditioning contributions through structural causal models. (c) Temporal importance re-weighting mechanism that transforms causal effects into normalized importance weights $\omega_P^g(t')$ and $\omega_{I_r}^g(t')$.

**Definition 1 (Step-wise Causal Intervention).** As shown in Figure 3 (b), a step-wise causal intervention at timestep $t'$ modifies only the transition at $t'$ while maintaining standard conditions elsewhere:

$$\text{do}(C = \varnothing, t'): \quad \boldsymbol{x}_{t-\Delta t} = \begin{cases} f_\theta(\boldsymbol{x}_t, \cdot, \cdot, \epsilon_t) \setminus C, & t = t' \\ f_\theta(\boldsymbol{x}_t, P, I_r, \epsilon_t), & t \neq t' \end{cases} \tag{5}$$

where $C \in \{P, I_r\}$ represents the ablated condition.

## 4.1 DECOUPLED CAUSAL INTERVENTION TRAJECTORIES

For comprehensive causal analysis, we generate three types of trajectories for each initial noise $\boldsymbol{x}_1^{(g)}$:

**Main trajectory**:

$$\{\boldsymbol{x}_t^{(g)}\}_{t=1}^{t=0}: \quad \boldsymbol{x}_{t-\Delta t}^{(g)} = f_\theta(\boldsymbol{x}_t^{(g)}, P, I_r, \epsilon_t) \tag{6}$$

**Prompt-intervention trajectory** at step $t'$:

$$\{\boldsymbol{x}_t^{(P,t',g)}\}_{t=1}^{t=0}: \quad \boldsymbol{x}_{t-\Delta t}^{(P,t',g)} = \begin{cases} f_\theta(\boldsymbol{x}_t^{(P,t',g)}, \varnothing, I_r, \epsilon_t), & t = t' \\ f_\theta(\boldsymbol{x}_t^{(P,t',g)}, P, I_r, \epsilon_t), & t \neq t' \end{cases} \tag{7}$$

**Reference-intervention trajectory** at step $t'$:

$$\{\boldsymbol{x}_t^{(I_r,t',g)}\}_{t=1}^{t=0}: \quad \boldsymbol{x}_{t-\Delta t}^{(I_r,t',g)} = \begin{cases} f_\theta(\boldsymbol{x}_t^{(I_r,t',g)}, P, \varnothing, \epsilon_t), & t = t' \\ f_\theta(\boldsymbol{x}_t^{(I_r,t',g)}, P, I_r, \epsilon_t), & t \neq t' \end{cases} \tag{8}$$

These intervention trajectories enable systematic analysis of how each conditioning signal influences generation at different temporal stages.

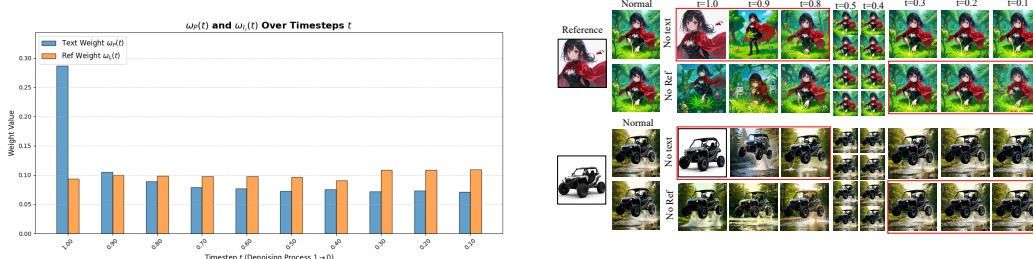

(a) **Temporal Causal Weights**  (b) **Step-wise Intervention**

Figure 4: **Causal Diagnostic Analysis.** (a) Statistical analysis reveals that text weights $\omega_P$ dominate early stages, while reference weights $\omega_{I_r}$ peak in late stages. (b) Visual interventions confirm this: early text ablation collapses structure, while late reference ablation degrades identity details.

### 4.1.1 QUANTIFYING TEMPORAL CAUSAL EFFECTS

We measure causal effects through specialized reward functions that evaluate different aspects of generation quality:

$$\mathcal{R}_P^{(g)} = \psi_P(\boldsymbol{x}_0^{(g)}, P), \quad \mathcal{R}_{I_r}^{(g)} = \psi_{I_r}(\boldsymbol{x}_0^{(g)}, I_r), \tag{9}$$

where $\psi_P$ measures text-image alignment and $\psi_{I_r}$ evaluates visual consistency.

We quantify the instantaneous causal contribution of each modality at timestep $t'$ by measuring the performance degradation resulting from its intervention, as shown in Figure 3 (c):

$$\delta_P^{(g)}(t') = \mathcal{R}_P^{(g)} - \psi_P(\boldsymbol{x}_0^{(P,t',g)}, P), \quad \delta_{I_r}^{(g)}(t') = \mathcal{R}_{I_r}^{(g)} - \psi_{I_r}(\boldsymbol{x}_0^{(I_r,t',g)}, I_r). \tag{10}$$

Higher values indicate stronger causal dependence on the conditioning signal at that timestep.

### 4.1.2 TEMPORAL IMPORTANCE RE-WEIGHTING

We convert causal effects into normalized importance weights that capture temporal dynamics:

$$\omega_P^{(g)}(t') = \frac{\exp(\delta_P^{(g)}(t')/\tau)}{\sum_t \exp(\delta_P^{(g)}(t)/\tau)}, \quad \omega_{I_r}^{(g)}(t') = \frac{\exp(\delta_{I_r}^{(g)}(t')/\tau)}{\sum_t \exp(\delta_{I_r}^{(g)}(t)/\tau)}, \tag{11}$$

where $\tau$ is a temperature parameter controlling the sharpness of the distribution. These weights explicitly reveal the temporal patterns of multi-modal influence during generation. We compute advantages that incorporate both temporal importance and group-level statistics:

**Key Observations.** These weights $\omega(t)$ explicitly quantify the shifting reliance between modalities, resolving the uniform timestep bias in standard GRPO. As illustrated in Figure 4, our causal diagnostics reveal a distinct **"Coarse-to-Fine" statistical law**:

- **Text Dominance (Early Steps):** In high-noise stages ($t \to 1$), the text weight $\omega_P(t)$ dominates. Visually, ablating the prompt here leads to structure collapse, confirming text drives global layout.
- **Reference Handover (Late Steps):** In low-noise stages ($t \to 0$), the reference weight $\omega_{I_r}(t)$ takes over. Ablating the reference here preserves structure but loses identity details, confirming visual signals anchor fine-grained textures.

By incorporating these weights into the advantage calculation (Eq. 17), we ensure gradients are amplified precisely at the critical decision points for each modality.

$$\mathcal{A}_P^{(g)}(t') = \omega_P^{(g)}(t') \cdot \frac{\mathcal{R}_P^{(g)} - \mu_P}{\sigma_P}, \quad \mathcal{A}_{I_r}^{(g)}(t') = \omega_{I_r}^{(g)}(t') \cdot \frac{\mathcal{R}_{I_r}^{(g)} - \mu_{I_r}}{\sigma_{I_r}}, \tag{12}$$

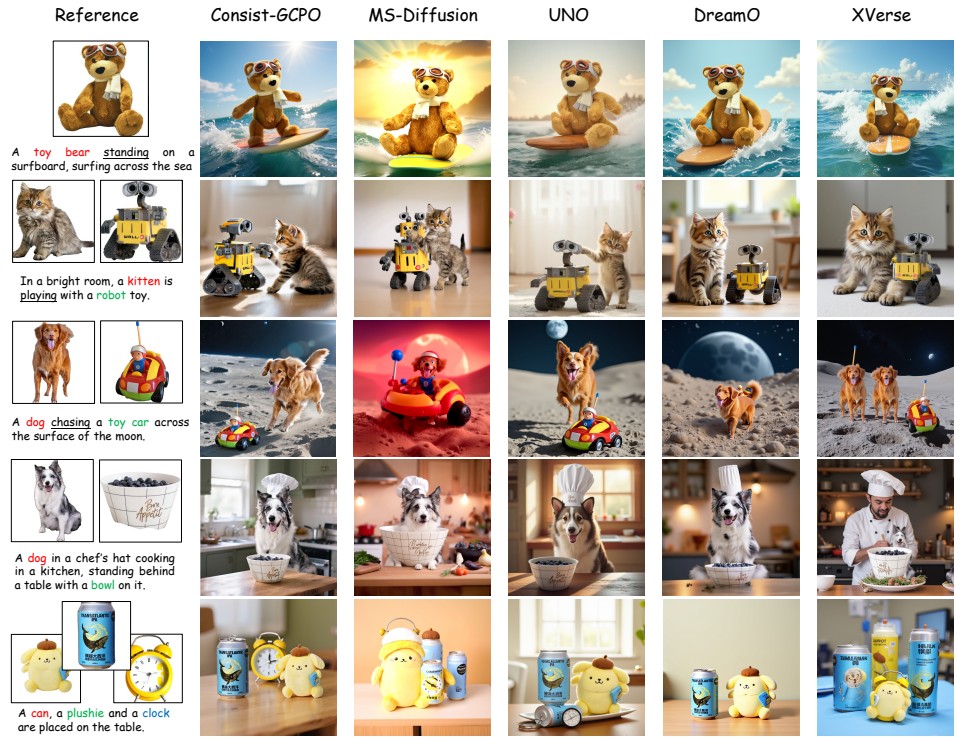

Figure 5: Qualitative comparison of image generation results. **Consis-GCPO** achieves state-of-the-art consistency and text fidelity in both single-reference and multi-reference image scenarios.

where $\mu_P = \text{Mean}[\{\mathcal{R}_P^{(j)}\}_{j=1}^G]$ and $\sigma_P = \text{Std}[\{\mathcal{R}_P^{(j)}\}_{j=1}^G]$ are group statistics.

The total advantage combines both modalities:

$$\mathcal{A}^{(g)}(t') = \lambda_P \mathcal{A}_P^{(g)}(t') + \lambda_{I_r} \mathcal{A}_{I_r}^{(g)}(t'), \tag{13}$$

where $\lambda_P$ and $\lambda_{I_r}$ are balancing coefficients that control the relative importance of each modality.

## 4.2 PROXIMAL POLICY OPTIMIZATION WITH CAUSAL WEIGHTING

The objective incorporates weighted advantages into a proximal policy optimization framework:

$$\mathcal{L}_{\text{Consis-GCPO}}(\theta) = -\frac{1}{G}\sum_{g=1}^G\sum_{t'}(min(r_t^g(\theta)\mathcal{A}^g(t'), \text{clip}(r_t^g(\theta), 1-\sigma, 1+\sigma)\mathcal{A}^g(t')) - \beta D_{KL}(\pi_\theta || \pi_{\text{ref}})), \tag{14}$$

where $r_{t'}^g = \frac{p_\theta(\boldsymbol{x}_{t'-\Delta t}^g | \boldsymbol{x}_{t'}^g)}{p_{\theta_{\text{old}}}(\boldsymbol{x}_{t'-\Delta t}^g | \boldsymbol{x}_{t'}^g)}$, $\sigma$ controls clipping for stability, and $\beta$ weights the KL regularization, keeping the same as Eqn 4. Although we aggregate objectives ($\mathcal{L} \propto \mathcal{A}_P + \mathcal{A}_{I_r}$), *disentanglement* is achieved at the **advantage estimation level**. Since $\mathcal{A}_P$ and $\mathcal{A}_{I_r}$ derive from independent counterfactuals (Eq. 10), their gradients are mathematically isolated. This formulation achieves targeted credit assignment by upweighting gradients at timesteps where each conditioning modality demonstrates high causal influence, leading to more efficient and interpretable multi-modal alignment.

## 5 EXPERIMENTS

### 5.1 EXPERIMENTAL SETUP

**Model Architecture and Reward Configuration.** We evaluate Consis-GCPO across image and video generation using a unified framework. For image synthesis, we employ UNO (Wu et al., 2025)

Table 1: Image quantitative comparison on DreamBench benchmark. The best results are in **bold**, and second-best are underlined. **Ours** includes standard deviation ($\pm$) over 5 runs. $^*$ indicates statistical significance ($p < 0.05$) against the best baseline.

| Method | Subject | CLIP-T $\uparrow$ | CLIP-I $\uparrow$ | DINO $\uparrow$ |
|---|---|---|---|---|
| MS-Diffusion (Wang et al., 2025) | Single | 0.311 | 0.808 | 0.703 |
| DreamO (Mou et al., 2025) | Single | 0.306 | 0.833 | 0.760 |
| XVerse (Chen et al., 2025) | Single | 0.302 | 0.832 | 0.754 |
| UNO (Wu et al., 2025) | Single | 0.304 | 0.835 | 0.760 |
| UNO + Flow-GRPO (Ruiz et al., 2023) | Single | 0.314 | 0.839 | 0.766 |
| UNO + Dance-GRPO (Li et al., 2023) | Single | 0.301 | 0.841 | 0.772 |
| **Consis-GCPO** | Single | **0.325**$\pm$0.003$^*$ | **0.848**$\pm$0.002$^*$ | **0.781**$\pm$0.004$^*$ |
| MS-Diffusion (Wang et al., 2025) | Multiple | 0.319 | 0.726 | 0.525 |
| DreamO (Mou et al., 2025) | Multiple | 0.321 | 0.733 | 0.522 |
| XVerse (Chen et al., 2025) | Multiple | 0.312 | 0.735 | 0.537 |
| UNO (Wu et al., 2025) | Multiple | 0.322 | 0.733 | 0.542 |
| UNO + Flow-GRPO (Wu et al., 2025) | Multiple | 0.325 | 0.742 | 0.551 |
| UNO + Dance-GRPO (Wu et al., 2025) | Multiple | 0.320 | 0.750 | 0.561 |
| **Consis-GCPO** | Multiple | **0.331**$\pm$0.004$^*$ | **0.772**$\pm$0.005$^*$ | **0.572**$\pm$0.004$^*$ |

for single-reference and multi-subject scenarios, while video generation uses Vace-1.3B (Jiang et al., 2025). For reward design, we implement domain-specific mechanisms targeting semantic alignment and visual consistency. For R2I generation, ImageReward (Xu et al., 2023) serves as text-image alignment evaluator $R_P$, while DINOv3 (Siméoni et al., 2025) computes visual similarity between reference and generated images as $R_{I_r}$. For R2V generation, VideoAlign (Liu et al., 2025b) provides text-video alignment assessment as $R_P$, and DINOv3 processes sampled frames (initial, middle, final) for efficient $R_{I_r}$ evaluation.

**Datasets and Benchmarks.** Training data combines Subject200K (Tan et al., 2024) and FFHQ (Karras et al., 2019), with GPT generating 5,000 diverse text-image pairs covering various semantic concepts and visual styles. We evaluate on DreamBench (Ruiz et al., 2023) for image generation and introduce Dream-VBench—extending DreamBench subjects with GPT-generated action prompts—yielding 500 video test samples (with reward ablation and model efficient in **Appendix**).

**Evaluation Protocol.** Our comprehensive framework spans multiple dimensions: CLIP-T measures semantic alignment through cosine similarity between CLIP (Radford et al., 2021) text and image embeddings; CLIP-I quantifies cross-modal consistency using CLIP visual features; DINO-I (Oquab et al., 2024) evaluates fine-grained visual similarity via ViT-S/16 features for identity preservation. For videos, we extend these to per-frame analysis and introduce Temporal Consistency, measuring inter-frame coherence through consecutive CLIP embedding similarities.

## 5.2 COMPARISON ON IMAGE CONSISTENCY GENERATION TASKS

We evaluate Consis-GCPO against state-of-the-art methods including MS-Diffusion (Wang et al., 2025), DreamO (Mou et al., 2025), XVerse (Chen et al., 2025), UNO (Wu et al., 2025), and GRPO variants on DreamBench. Table 1 demonstrates our method achieves superior performance across all metrics in both single-subject and multi-subject scenarios, with particularly pronounced improvements in complex multi-reference conditioning—obtaining CLIP-T (0.331), CLIP-I (0.772), and DINO (0.572) for multi-subject generation. Figure 5 provides qualitative validation, showing Consis-GCPO's excellence in instruction adherence while maintaining subject consistency. **Notably,** only our method correctly generates the "standing" posture for the teddy bear rather than replicating the sitting reference pose, and uniquely captures the dynamic "chasing" behavior between dog and toy car as specified in prompts. These results demonstrate Consis-GCPO's capability for subject fidelity with precise text-following.

## 5.3 EVALUATION ON VIDEO CONSISTENCY.

We compare **Consis-GCPO** against recent leading video generation frameworks, including VACE (Jiang et al., 2025), MAGREF (Deng et al., 2025), Phantom (Liu et al., 2025c), and Video-Booth (Jiang et al., 2024). To provide a more comprehensive analysis, we also include comparisons with two reinforcement learning–based optimization strategies: Flow-GRPO (Liu et al., 2025a) and

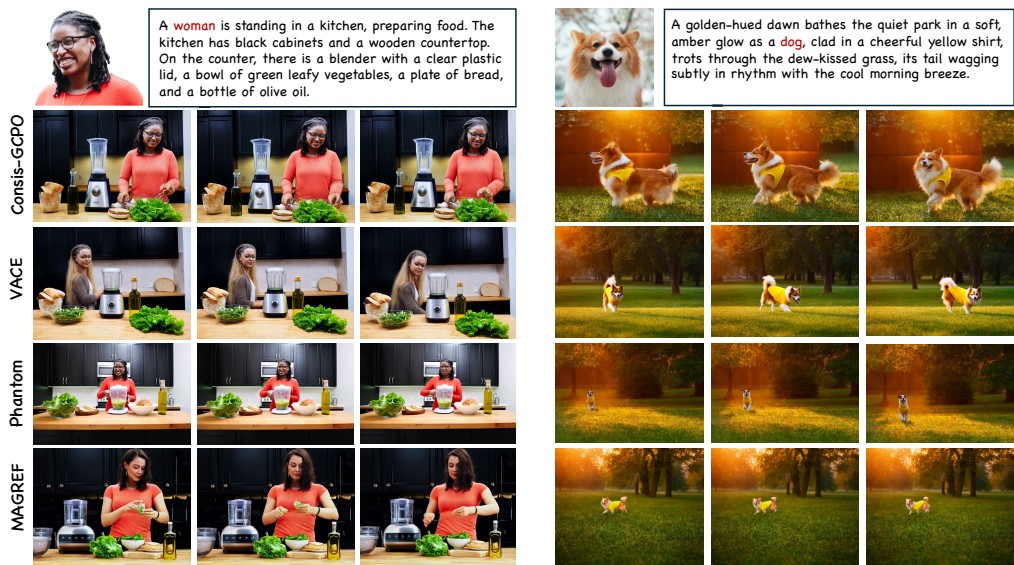

Figure 6: Qualitative comparison of video generation results. **Consis-GCPO** achieves superior subject consistency and temporal coherence while maintaining precise text alignment, demonstrating robust performance across diverse multi-subject scenarios with complex motions and interactions.

Table 2: Video quantitative comparison for single- and multi-subject on Dream-VBench benchmark. The best results are in **bold**, and second-best are underlined. **Consis-GCPO** includes standard deviation ($\pm$) over 5 runs. * indicates statistical significance ($p < 0.05$) against the best baseline.

| Methods | Subject | CLIP-T ↑ | CLIP-I ↑ | DINO-I ↑ | Consistency ↑ |
|---|---|---|---|---|---|
| VideoBooth (Jiang et al., 2024) | Single | 0.267 | 0.523 | 0.634 | 0.938 |
| MAGREF (Deng et al., 2025) | Single | 0.278 | 0.669 | 0.675 | 0.965 |
| Phantom-1.3B (Liu et al., 2025c) | Single | 0.266 | 0.601 | 0.710 | 0.963 |
| Vace-1.3B (Jiang et al., 2025) | Single | 0.277 | 0.727 | 0.697 | 0.970 |
| Vace-1.3B + Flow-GRPO (Liu et al., 2025a) | Single | 0.271 | 0.759 | 0.719 | 0.978 |
| Vace-1.3B + DanceGRPO (Xue et al., 2025) | Single | 0.287 | 0.755 | 0.732 | 0.981 |
| **Consis-GCPO** | Single | **0.305**$_{\pm0.004}$* | **0.790**$_{\pm0.003}$* | **0.746**$_{\pm0.003}$* | **0.984**$_{\pm0.001}$* |
| Vace-1.3B (Jiang et al., 2025) | Multiple | 0.274 | 0.615 | 0.589 | 0.966 |
| Vace-1.3B + Flow-GRPO (Liu et al., 2025a) | Multiple | 0.265 | 0.645 | 0.587 | 0.974 |
| Vace-1.3B + DanceGRPO (Xue et al., 2025) | Multiple | 0.281 | 0.642 | 0.594 | 0.972 |
| **Consis-GCPO** | Multiple | **0.300**$_{\pm0.005}$* | **0.674**$_{\pm0.005}$* | **0.608**$_{\pm0.007}$* | **0.981**$_{\pm0.002}$* |

DanceGRPO (Xue et al., 2025). The evaluation considers three dimensions central to reference-guided video generation: (i) *reference-video visual fidelity*, (ii) *text-video semantic alignment*, and (iii) *temporal consistency*. Figure 6 illustrates these improvements qualitatively. Quantitative results are summarized in Table 2. Our method achieves consistent improvements across all settings. In semantic alignment, *Consis-GCPO* attains a CLIP-T score of 0.305, which surpasses the strongest baseline (VACE+DanceGRPO: 0.287) by 6.3%. For identity preservation, our DINO-I score of 0.746 exceeds the prior best (0.732), highlighting stronger reference fidelity. Most notably, our temporal consistency reaches 0.984, outperforming all variants including Flow-GRPO and Dance-GRPO, whose best results plateau at 0.978 and 0.981, respectively.

## 5.4 ANALYSIS ON STEP-WISE COUNTERFACTUAL INTERVENTIONS

To assess our decoupled causal intervention mechanism, we conduct ablation studies examining four configurations: (1) baseline Flow-GRPO without interventions, (2) prompt-only interventions $do(P = \emptyset, t')$, (3) reference-only interventions $do(I_r = \emptyset, t')$, and (4) our complete framework. Table 3 presents quantitative results for image and video generation. For multi-subject image generation, baseline Flow-GRPO yields suboptimal performance (CLIP-T: 0.325, DINO-I: 0.551), demonstrating limitations of uniform temporal optimization. Prompt-only intervention improves text align-

Table 3: Ablation study on causal interventions in multi-subject R2I and R2V generation scenarios.

| Method | R2I Generation | | | R2V Generation | | |
|---|---|---|---|---|---|---|
| | CLIP-T ↑ | CLIP-I ↑ | DINO-I ↑ | CLIP-T ↑ | CLIP-I ↑ | DINO-I ↑ |
| No Interventions (Flow-GRPO) | 0.325 | 0.742 | 0.551 | 0.265 | 0.645 | 0.587 |
| Prompt-only Interventions | **0.338** | 0.736 | 0.544 | **0.310** | 0.628 | 0.556 |
| Reference-only Interventions | 0.322 | **0.780** | 0.570 | 0.255 | 0.670 | **0.615** |
| Full Interventions (Ours) | 0.331 | 0.772 | **0.572** | 0.300 | **0.674** | 0.608 |

ment (CLIP-T: 0.338) but shows marginal identity preservation gains (DINO-I: 0.544), indicating temporal credit assignment for textual conditioning alone is insufficient. Reference-only intervention significantly enhances visual consistency (CLIP-I: 0.780) while maintaining reasonable text alignment (CLIP-T: 0.322). Our complete framework outperforms all partial configurations (CLIP-T: 0.331, DINO-I: 0.572), empirically validating that independent temporal credit assignment for both conditioning modalities is essential for optimal subject-driven generation performance.

## 5.5 COMPARISON OF OPTIMIZATION STRATEGIES

To validate the design rationale discussed in Section 4.2, we compare our Joint Optimization against two baselines: **Alternating Optimization** (updating text and image rewards alternately every 2 steps) and **Sequential Optimization** (optimizing text first for 50% steps, then visual consistency).

As shown in Table 4, the Joint strategy yields superior performance and efficiency. The Alternating method suffers from gradient oscillation, resulting in suboptimal convergence (DINO-I: 0.762). The Sequential approach exhibits catastrophic forgetting, evidenced by the sharp drop in text alignment (CLIP-T: 0.308) during the second phase. Furthermore, Joint Optimization is **1.8× faster** than alternating methods by sharing the backward pass, empirically confirming it as the Pareto-optimal choice for stability and training cost.

Table 4: **Ablation on optimization strategies.** Joint optimization achieves superior stability and efficiency compared to alternating or sequential methods.

| Strategy | CLIP-T ↑ | CLIP-I ↑ | DINO-I ↑ | Efficiency |
|---|---|---|---|---|
| Alternating (Text ↔ Image) | 0.317 | 0.837 | 0.762 | 1.8× (Slower) |
| Sequential (Text → Image) | 0.308 | 0.842 | 0.770 | 1.5× (Slower) |
| **Ours (Joint)** | **0.325** | **0.848** | **0.781** | **1.0×** |

## 6 CONCLUSION

We presented Consis-GCPO, a causal reinforcement learning framework that addresses fundamental limitations in multimodal personalized generation. Through step-wise causal interventions, we enable precise quantification of when textual semantics versus visual references are most critical during denoising. Through temporally-weighted advantage computation, we transform instantaneous causal effects into targeted optimization signals that decouple text and reference contributions. Our experiments demonstrate that Consis-GCPO achieves superior subject consistency while preserving strong text-following capabilities, particularly excelling in complex multi-subject scenarios.

**Limitations.** While our method achieves significant improvements, current experiments focus on algorithmic innovations rather than reward model enhancements. **Future work** will explore incorporating multi-modal rewards from more powerful foundation models to develop comprehensive reward frameworks for enhanced multi-dimensional performance.

**Use of LLMs.** We utilize LLMs to assist with formula derivations and writing refinement.

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

# A APPENDIX

## A.1 ADDITIONAL IMPLEMENTATION DETAILS

### A.1.1 DATA PREPROCESSING PIPELINE

To ensure reproducibility and consistency with standard community protocols, we adopted a minimal preprocessing pipeline:

- **Visual Preprocessing:** All images were resized such that the shortest side is 512, followed by a center crop to $512 \times 512$ resolution. Pixel values were normalized to the range $[-1, 1]$. No random augmentations (e.g., flipping, rotation) were applied during fine-tuning to maintain precise alignment between the prompt and visual content.
- **Text Preprocessing:** Text prompts were tokenized using the standard CLIP tokenizer with a context length of 77 tokens. Sequences exceeding this limit were truncated, and shorter ones were padded, consistent with the pre-trained text encoder's requirements.

### A.1.2 VIDEO DATA CONSTRUCTION PIPELINE

For video generation experiments, we constructed a specialized dataset of 5,000 motion-aware text-image pairs. We utilized the same visual references from Subject200K and FFHQ but synthesized dynamic temporal instructions. Representative examples demonstrating the dataset quality are shown in Figure 7.

- **Motion-Aware Prompt Generation:** We employed an automated pipeline using GPT-4 to act as a "Video Director." It converted static image captions into dynamic scripts by injecting temporal predicates (e.g., "turning," "running") and cinematic instructions (e.g., "zoom in," "pan right").
- **Filtering Criteria:** Generated prompts were filtered based on strict inclusion criteria: (1) Must contain at least one dynamic verb; (2) Length constraint of 20-50 words for conciseness; (3) Semantic consistency with the reference object class.
- **Example:** *Static Input:* "A girl smiling." → *Dynamic Output:* "Cinematic shot of a girl slowly breaking into a warm smile, wind blowing her hair, 4k detail."

| Reference Image | Source Dataset | R2I Caption | R2V Caption |
|---|---|---|---|
| | Subject200K | A parked military helicopter with a desert camo scheme, sitting on a dusty tarmac under bright daylight, detailed mechanical textures. | A dramatic aerial view of the military helicopter banking sharply through a misty mountain valley. The side door is open as it maneuvers past rocky cliffs, with pine trees rushing by below. The camera follows the aircraft as it accelerates towards the horizon under a cloudy sky. |
| | Subject200K | A detailed full-body photograph of the penguin standing upright on an icy surface with a blurred Antarctic snow background. | An underwater shot showing the penguin diving from an ice shelf into clear blue water. It swims gracefully using its flippers, leaving a trail of bubbles behind it, as light filters down from the surface, illuminating the icy environment. |
| | FFHQ | A medium-shot portrait of a woman standing in a blooming garden, holding a woven basket full of colorful flowers. She is facing the camera with a bright smile, bathed in warm afternoon sunlight. | A medium shot as the woman slowly turns her head from looking off-camera to directly facing the lens. She smiles gently and nods her head once, with sunlight filtering through a window, catching her hair. |

Figure 7: **Data samples illustrating our task-specific caption construction**. We show representative reference images from different source datasets (FFHQ, Subject200K) alongside their corresponding textual conditions. The table contrasts the Static R2I Captions (focusing on visual appearance) with the Motion-Aware R2V Captions (incorporating specific temporal dynamics and cinematic instructions).

# B  ADDITIONAL ANALYSIS

## B.1  SENSITIVITY ANALYSIS ON HYPERPARAMETERS

We conducted extensive ablation studies to evaluate the robustness of our method to key hyperparameters.

### B.1.1  BALANCING COEFFICIENTS ($\lambda_P, \lambda_{I_r}$).

We fixed $\lambda_P = 1.0$ and varied $\lambda_{I_r}$. As shown in Table 5, the balanced setting ($\lambda_P = \lambda_{I_r} = 1.0$) achieves the optimal trade-off. Increasing $\lambda_{I_r}$ yields diminishing returns in identity metrics while significantly degrading text alignment.

Table 5: **Impact of balancing coefficients.** We report CLIP-I, CLIP-T and DINO-I to explicitly evaluate visual quality.

| $\lambda_P$ | $\lambda_{I_r}$ | CLIP-T ↑ | CLIP-I ↑ | DINO-I ↑ | Avg. ↑ |
|---|---|---|---|---|---|
| 1.0 | 0.5 | 0.332 | 0.835 | 0.760 | 0.642 |
| **1.0** | **1.0** | **0.325** | **0.848** | **0.781** | **0.651** |
| 1.0 | 1.2 | 0.318 | 0.850 | 0.783 | 0.650 |
| 1.0 | 2.0 | 0.305 | 0.852 | 0.788 | 0.648 |

### B.1.2  TEMPERATURE PARAMETER ($\tau$).

Table 6 presents the quantitative ablation on the softmax temperature $\tau$. The results confirm that $\tau = 1.0$ provides the optimal balance between temporal specialization and gradient density. Extreme values degrade performance: $\tau = 0.8$ yields overly sparse gradients, while $\tau = 1.2$ leads to uniform weighting that mimics the baseline GRPO.

**Visualization and Modality-Specific Analysis.** To understand the underlying mechanism, we visualize the learned temporal weight curves under different $\tau$ settings in Figure 8.

- **Impact of High Temperature ($\tau = 1.2$):** Increasing $\tau$ overly smoothes the distribution. As shown in the visualization, the distinct peak of reference guidance at late stages is flattened, causing the text weight to remain relatively high even when it should yield control. This **loss of modal discrimination** prevents the model from focusing exclusively on identity refinement, degrading DINO-I scores.

- **Impact of Low Temperature ($\tau = 0.8$):** Conversely, decreasing $\tau$ makes the distribution overly sharp. While it highlights the peak steps, it forces the weights of adjacent supportive steps to near zero. This **gradient sparsity** means the model receives no optimization signal for valid transitional timesteps, leading to unstable training and a drop in overall performance.

## B.2  ANALYSIS ON REWARD MODEL

We further investigate the impact of different reward models on video and image generation under the single-reference setting. As shown in Table 7, reward signals are divided into text-related ($R_P$) for semantic alignment and reference-related ($R_{I_r}$) for identity preservation. For $R_P$, VideoAlign achieves the highest average performance in video generation (0.614), whereas ImageReward is

Table 6: **Impact of temperature $\tau$.** Extreme values (too sharp or too flat) degrade performance.

| $\tau$ | Type | CLIP-T ↑ | CLIP-I ↑ | DINO-I ↑ | Avg. ↑ |
|---|---|---|---|---|---|
| 0.8 | Sharp | 0.316 | 0.839 | 0.765 | 0.640 |
| **1.0** | **Ours** | **0.325** | **0.848** | **0.781** | **0.651** |
| 1.2 | Flat | 0.319 | 0.845 | 0.772 | 0.645 |

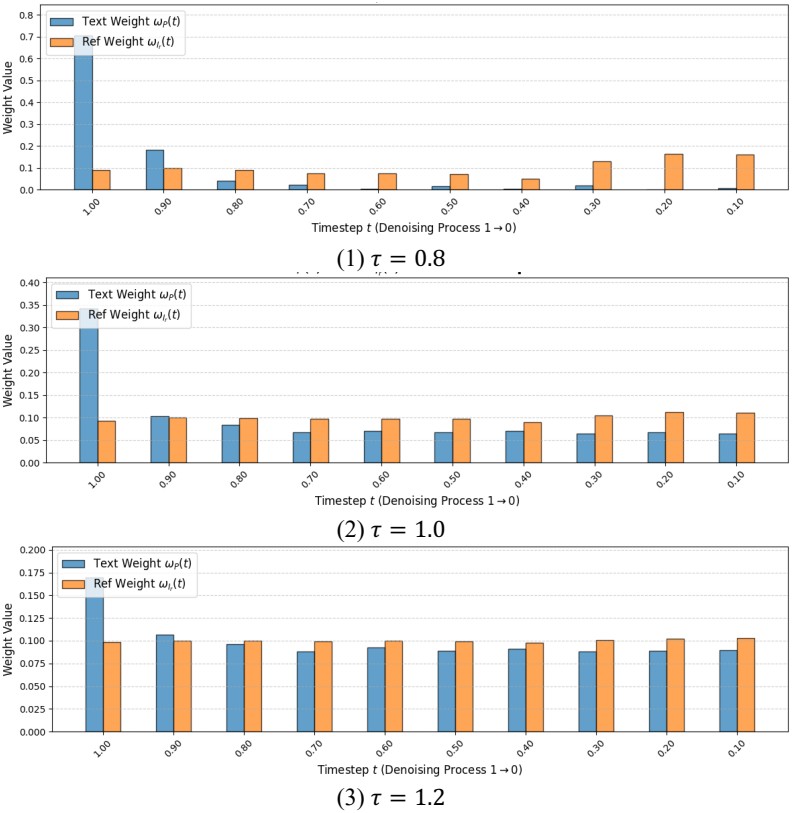

Figure 8: **Visualization of causal weights under different temperatures** $\tau$. $\tau = 1.0$ (Ours) achieves a clear separation of modal influence. $\tau = 1.2$ leads to over-smoothing (leakage), while $\tau = 0.8$ leads to over-sharpening (information loss).

superior in image generation (0.651). CLIP-T attains the best CLIP-T score for video (0.322) but lags on other metrics. For $R_{I_r}$, DINOv3 consistently outperforms CLIP-I and DINOv2, obtaining the highest averages in both video (0.614) and image (0.651). Therefore, we adopt VideoAlign for video generation and ImageReward for image generation as $R_P$, together with DINOv3 as $R_{I_r}$, since this combination provides the most favorable trade-off between semantic alignment and identity consistency.

Table 7: Ablation on reward models for video and image generation. **For clarity we fix the best-performing $R_P$ (or $R_{I_r}$) and report comparisons by varying the other reward model.** The best results are in **bold**, and the second-best are underlined.

| Method | R2I Generation | | | | R2V Generation | | | |
|---|---|---|---|---|---|---|---|---|
| | CLIP-T ↑ | CLIP-I ↑ | DINO-I ↑ | Avg. ↑ | CLIP-T ↑ | CLIP-I ↑ | DINO-I ↑ | Avg. ↑ |
| $R_P$ (with $R_{I_r}$ = DINOv3) | | | | | | | | |
| Qwen2.5-VL (Qwen et al., 2025) | 0.303 | 0.779 | 0.727 | 0.603 | 0.297 | 0.738 | 0.704 | 0.580 |
| CLIP-T Radford et al. (2021) | 0.324 | 0.710 | 0.700 | 0.578 | **0.322** | 0.723 | 0.711 | 0.585 |
| ImageReward (Xu et al., 2023) | **0.325** | **0.848** | **0.781** | **0.651** | - | - | - | - |
| VideoAlign Liu et al. (2025b) | - | - | - | - | 0.305 | **0.790** | **0.746** | **0.614** |
| $R_{I_r}$ (with $R_p$ = ImageReward for R2I and $R_p$ = VideoAlign for R2V) | | | | | | | | |
| CLIP-I Radford et al. (2021) | 0.280 | **0.861** | 0.694 | 0.612 | 0.279 | **0.795** | 0.691 | 0.588 |
| DINOv2 (Oquab et al., 2024) | 0.291 | 0.829 | **0.786** | 0.635 | 0.288 | 0.720 | **0.782** | 0.597 |
| DINOv3 Siméoni et al. (2025) | **0.325** | 0.848 | 0.781 | **0.651** | **0.305** | 0.790 | 0.746 | **0.614** |

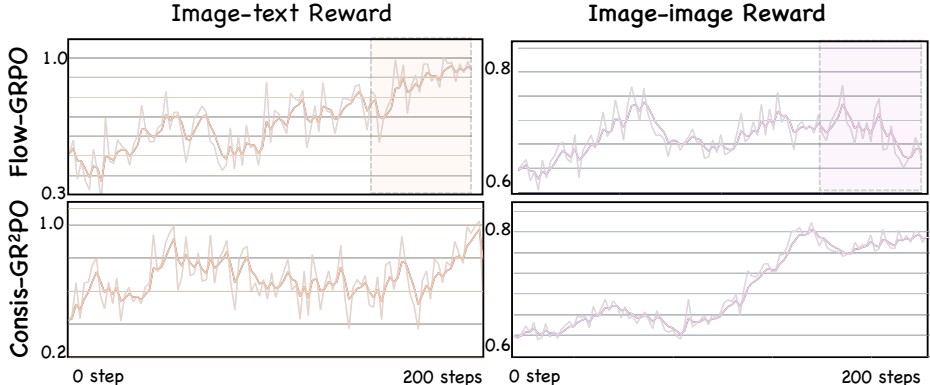

Figure 9: **Training reward trajectories of FlowGRPO and Consis-GCPO.**

### B.3 ANALYSIS OF MODEL EFFICIENCY

We assess the efficiency of the proposed framework by analyzing both computational cost and reward evolution, as illustrated in Figure 9 and Figure 10.

#### B.3.1 TRAINING REWARD TRAJECTORIES.

As shown in Figure 9, we examine the trajectories of text-related and reference-related rewards during training for *FlowGRPO* and *Consis-GRPO*. Our results show that, in later training stages, the strong semantic alignment characteristic of FlowGRPO constrains consistency improvements and even introduces detrimental side effects. In contrast, Consis-GRPO effectively circumvents these issues by decoupling semantic adherence from consistency optimization. Furthermore, under an identical training budget of 200 steps, Consis-GRPO achieves considerably higher reference-related rewards compared to FlowGRPO, indicating superior consistency in generation.

#### B.3.2 COMPUTATIONAL EFFICIENCY AND CONVERGENCE ANALYSIS

To investigate the benefits of our timestep-aware optimization, we provide a comprehensive analysis of the trade-off between per-step computational cost and overall convergence efficiency.

**Mechanism: Inference vs. Training Cost.** While our method involves calculating counterfactual trajectories, these additional rollouts are performed in `no_grad` mode (inference only). The computationally expensive backpropagation is performed *only* on the main trajectory. Therefore, the gradient computation overhead remains comparable to standard fine-tuning. Our method effectively trades "cheap" inference compute for "expensive" training iterations.

**Sample Efficiency and Total Convergence Time.** By extracting high-quality causal gradients, Consis-GCPO achieves a dramatic improvement in sample efficiency compared to the baseline:

- **Baseline (Flow-GRPO):** Requires $\approx 15,000$ steps to converge.
- **Ours (Consis-GCPO):** Converges in $\approx 1,300$ steps.

Despite the increased inference load per step (approximately $8\times$ for trajectory rollout), this $11.5\times$ reduction in required training steps results in a significant net gain.

**Wall-Clock Speedup.** We further analyze the evaluation rewards normalized by GPU hours to measure real-world efficiency. As presented in Figure 10, Consis-GCPO attains the target evaluation reward approximately **1.4× faster** than Flow-GRPO in terms of wall-clock time. This demonstrates that the proposed causal intervention accelerates convergence without compromising performance, validating our design choice of investing in causal gradient quality.

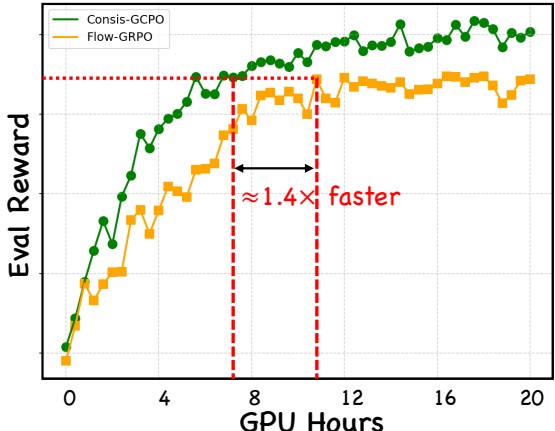

Figure 10: **Evaluation rewards vs. GPU hours.** Consis-GRPO reaches the same reward level about $1.4\times$ faster than FlowGRPO, highlighting its improved computational efficiency.

## C    MORE VISUALIZATION

### C.1    PROMPTS IN FIGURE 1.

The prompts in Figure 1 are as follows:

---

1. In a park, A man is holding a camera shotting.
2. A cartoon character chasing a toy car across the surface of the moon, under natural daylight, with realistic details of lunar soil, craters, and the vast space backdrop.
3. An anime space ranger is riding a bicycle in front of a cyberpunk skyscraper background.
4. A cinematic scene of a man wearing a t-shirt outdoors petting a dog, warm sunlight, detailed textures, natural background.
5. The anime Spider-Man leaps across skyscrapers, clutching a roll of film.
6. A cinematic 4K video of a young woman gracefully playing the violin in a grand, opulent concert hall adorned with golden chandeliers and ornate decorations. The camera captures multiple angles in smooth motion: close-up shots of her hands moving the bow across the strings, mid-shots of her calm and focused expression, and wide shots revealing the majestic hall with its glittering lights and luxurious atmosphere.
7. A cute small dog wearing a red Christmas hat lies cozily on a fluffy rug in front of a glowing fireplace. The warm firelight flickers softly on the walls, creating a festive and comforting holiday atmosphere. After a moment of resting, the dog slowly gets up, shakes its body gently, and begins to walk forward.

---

### C.2    FAILURE CASES

We have identified two primary limitations where our method faces challenges, as visualized in Figure 11:

- **Extreme Semantic Conflict:** When the text prompt and visual reference are fundamentally contradictory (e.g., Prompt: "A cat", Reference: [Image of a dog] ), the causal reweighting mechanism struggles to reconcile the divergence. High causal effects are detected for *both* conflicting modalities simultaneously, which confuses the optimization and often results in hybrid artifacts or semantic oscillation.

- **Micro-Detail Loss:** In scenarios involving extremely small subjects within wide-angle shots (e.g., a tiny face in a crowd), the underlying reward models (DINO/CLIP) sometimes fail to capture identity loss accurately due to resolution limits. Consequently, even if our

method correctly upweights the relevant timesteps, the *reward signal itself* is too noisy to guide the recovery of micro-details.

Failure Case 1 : text prompt and visual reference are fundamentally contradictory.

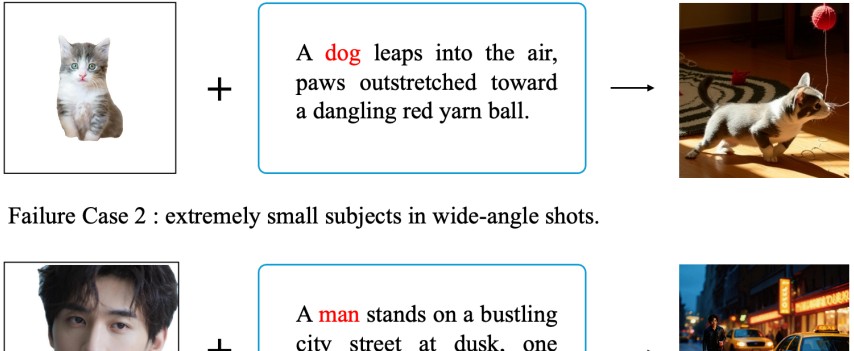

Failure Case 2 : extremely small subjects in wide-angle shots.

Figure 11: **Visualization of failure cases.** (Left) Hybrid artifacts resulting from extreme semantic conflict between prompt and reference. (Right) Loss of micro-details in wide-angle shots due to sparse reward signals.

## C.3 ADDITIONAL RESULTS

We further provide supplementary results on consistency generation. Figure 12 and Figure 13 presents examples of image-level consistency, while Figure 14 illustrates video-level consistency, demonstrating that our Consis-GCPO maintains coherent and stable outputs across both modalities.

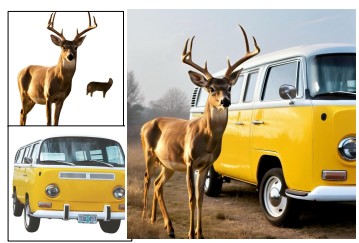

A deer standing beside a vintage van.

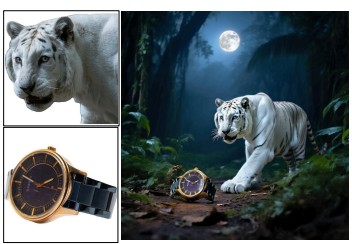

In the moonlit jungle, a white tiger prowls silently, its eye catching the glint of an old, abandoned watch nearby.

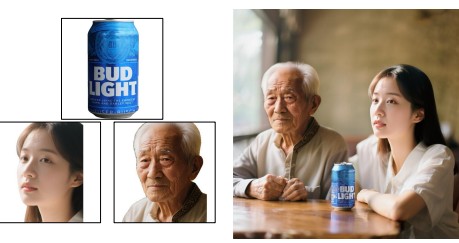

A woman and an old man are sitting together, with a beer can on the table between them.

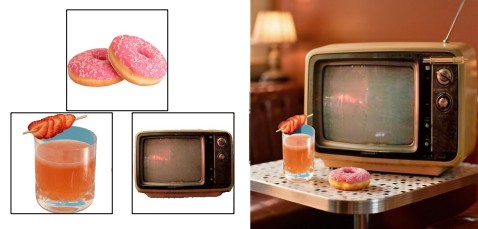

A vintage television is on, and there's a cocktail and a donut on the table beside it.

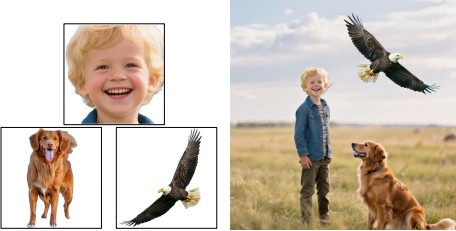

A boy is standing in an open field, with an eagle soaring in the sky above him and a dog sitting at his feet.

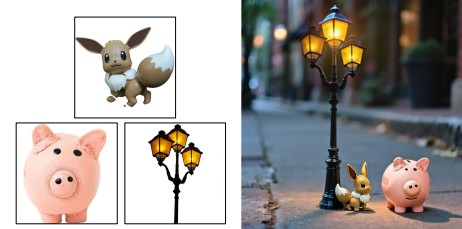

A street lamp illuminates an Eevee figurine placed next to a piggy bank on the sidewalk.

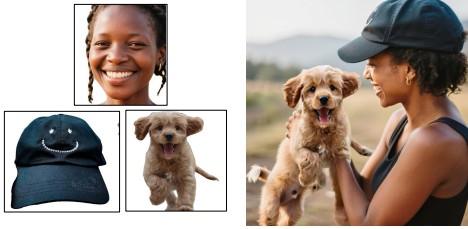

A woman with a cap playing with a puppy.

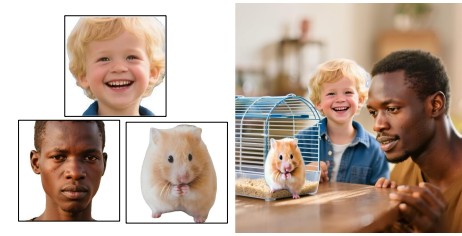

A boy and a man are watching a hamster play in a cage.

Figure 12: **More R2I Generation results.**

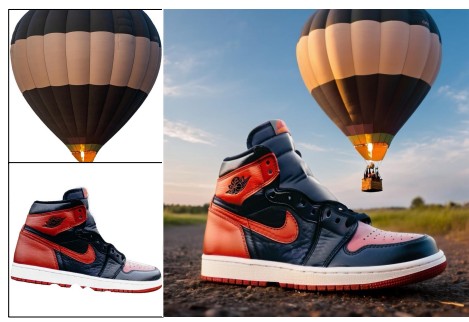

A hot air balloon floating in the sky above a sneaker lying on the ground.

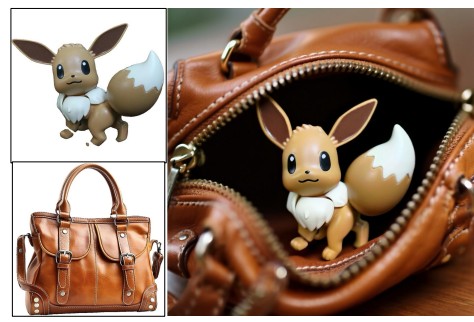

An Eevee figurine placed inside a leather handbag.

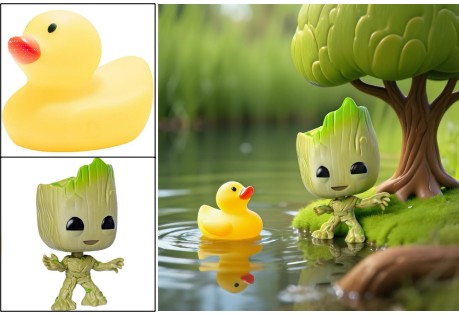

A rubber duck floating near a tree-like character by a small pond.

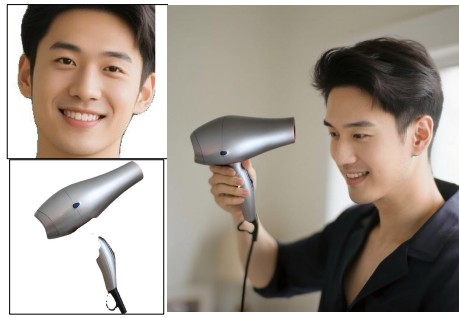

A man is using a hair dryer.

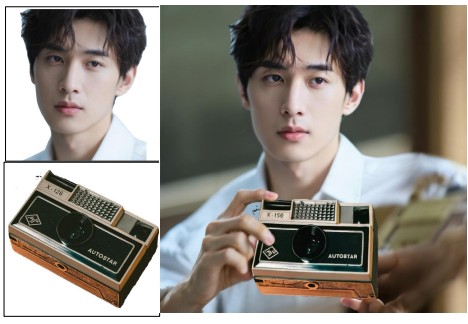

A man is holding a vintage camera.

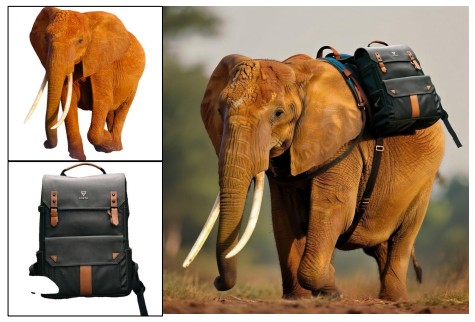

An elephant is carrying a backpack on its back.

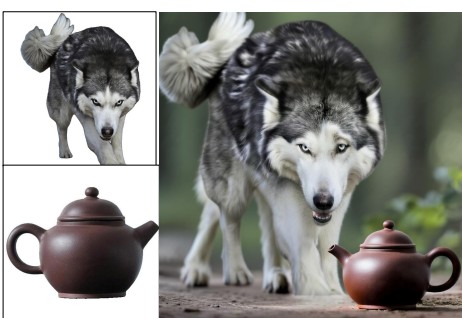

A wolf is standing beside a teapot.

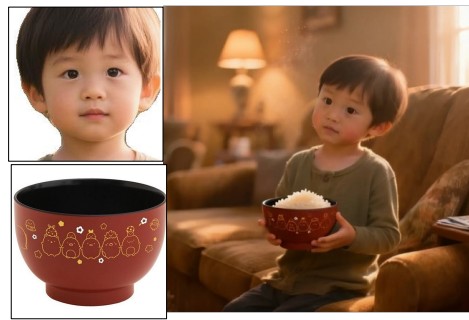

A boy holding a bowl filled with rice.

Figure 13: **More R2I Generation results.**

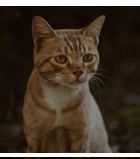
A small cat dressed in a flowing purple wizard robe sits at a wooden table under the soft glow of candlelight. The cat gazes intently at an ancient open book, its eyes reflecting the flickering flames. After a brief pause, the cat gently raises its paw and turns a page,

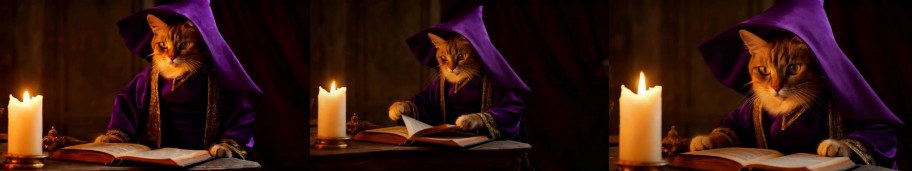

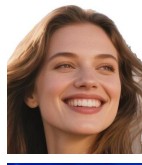
The video begins with a close-up of a woman passionately singing on a brightly lit stage, her hand gripping the microphone. She is wearing a shimmering silver sequin slip dress that sparkles under the intensified stage lights. The camera slowly pulls back, revealing colorful spotlights.

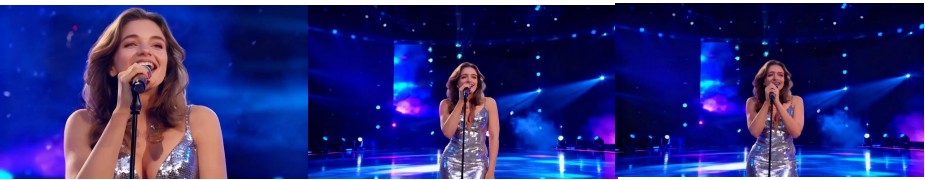

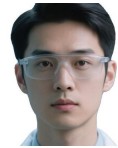
The video shows a man speaking to reporters in a locker room. He is wearing a blue shirt and appears to be addressing the media. The background features shelves filled with various sports equipment, including hockey gear such as helmets and gloves.

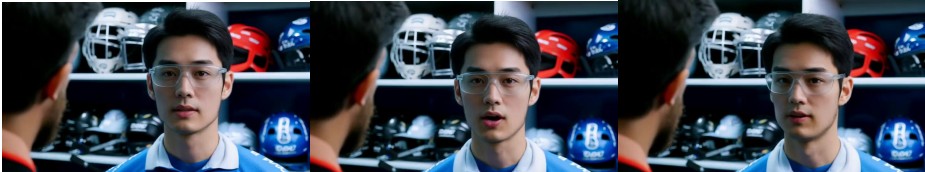

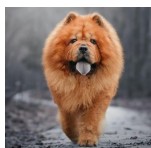
"A golden retriever sits calmly its fur gently ruffled by a cool morning breeze, while behind it a quaint blue house stands washed in the soft, golden light of sunrise, its windows reflecting faint glimmers of the awakening day."

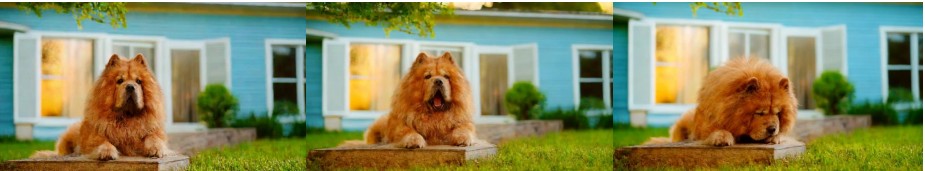

Figure 14: **More R2V Generation results.**

