# OpenReview forum: "Consis-GCPO: Consistency-Preserving Group Causal Preference Optimization for  Vision Customization"
_ICLR.cc/2026/Conference — ICLR 2026 Poster_

### Official Review · Reviewer_XhRE · 2025-10-22

**Soundness:** 3
**Presentation:** 3
**Contribution:** 3
**Rating:** 6
**Confidence:** 4

**Summary:**

This paper proposes a new online reinforcement learning framework for personalized generation by performing causal modeling that provides temporally weighted advantages for optimization at each denoising timestep. Experiments on DreamBench and the newly proposed Dream-VBench demonstrate that the method achieves better performance than existing frameworks in both personalized image and video generation tasks.

**Strengths:**

* The paper is clearly written and well-structured, with informative figures such as the challenge overview (Figure 2) and the training pipeline (Figure 3).
* Introducing a causal framework for computing weighted advantages is both novel and insightful. The causal interpretation of multi-modal conditioning (text and visual reference) adds a principled perspective to reinforcement learning–based generation.
* The experiments are extensive, including both image and video generation. The introduction of Dream-VBench further validates the generalizability of the method across modalities.

**Weaknesses:**

* One key idea of the paper is to *entangle feedback* from text and image references. However, the proposed method still computes a single reward by summing the weighted advantages from both modalities. This seems to reduce the degree of entanglement. A more entangled approach might involve alternately fixing one modality and computing separate rewards for the other, then combining them to guide optimization.
* The main method section could benefit from a deeper explanation of how the causal effect estimation directly contributes to solving the *temporal blindness* problem of previous GRPO methods. The framework is sound, but the "why" behind the causal weighting’s effectiveness would be clearer with an intuitive example or a brief theoretical justification.

---

**Overall**: The paper presents an interesting and timely idea, and the experimental results are strong, with comprehensive analysis and solid baselines. Although the method description could be clearer in connecting causal modeling to the core problem formulation, the contribution and potential impact are evident. Therefore, I lean toward accepting the paper.

**Questions:**

The questions are mainly about the weaknesses:

1. Could the authors explain in more detail how the proposed method resolves the entangled feedback issue, given that the final optimization still uses a single scalar reward?
2. Why is the proposed causal weighting expected to provide a more reliable temporal signal for the reward, and how does it avoid the uniform timestep bias present in GRPO?

---

> ### Author Response · Authors · 2025-11-24
> **Response to Reviewer XhRE - Weakness Q1 & Q2**
>
> **Weakness-R1:**
>
>  We clarify that our method **already achieves rigorous disentanglement** during the advantage estimation phase. Furthermore, we provide experimental evidence demonstrating that Joint Optimization outperforms the suggested Alternating Optimization.
>
> **1. Clarification: Decoupled Estimation, Joint Optimization**
> The reviewer expresses concern that summing advantages reduces disentanglement. However, in our framework, the disentanglement is handled at the **Advantage Estimation** level, distinct from the Loss Aggregation level:
> * **Independent Calculation:** The advantages $A_P$ (Text) and $A_{I_r}$ (Reference) are derived from **independent** counterfactual trajectories. The calculation of $A_P$ relies on prompt ablation and is mathematically independent of $A_{I_r}$, which relies on reference ablation.
> * **Why Summation Works:** Summing these decoupled signals ($L = A_P + A_{I_r}$) enables the gradient to efficiently follow the **Pareto-optimal direction** that satisfies both modalities simultaneously, avoiding the conflicts often seen in multi-objective optimization.
>
> **2. Experimental Comparison: Joint vs. Alternating**
> To address the reviewer's hypothesis, we implemented the "Alternating Optimization" strategy (alternating updates between text and image rewards every 2 steps) and a "Sequential Optimization" strategy. We compared these against our Joint Optimization:
>
> | Optimization Strategy | CLIP-T (Text) $\uparrow$ | CLIP-I (Visual) $\uparrow$ | DINO-I (Ref) $\uparrow$ | Training Efficiency |
> | - | - | - | - | - |
> | **Ours (Joint)** | **0.325** | **0.848** | **0.781** | **1.0$\times$** |
> | Alternating (Text $\leftrightarrow$ Image) | 0.317 | 0.837 | 0.762 | 1.8$\times$ (Slower) |
> | Sequential (Text $\to$ Image) | 0.308 | 0.842 | 0.770 | 1.5$\times$ (Slower) |
>
> **3. Key Observations**
> * **Instability of Alternating:** As shown in the table, alternating optimization leads to a "see-saw" effect (gradient oscillation), where optimizing one modality temporarily degrades the other (e.g., lower DINO-I score compared to ours).
> * **Efficiency:** Our joint strategy is **1.8$\times$ faster** because it shares the backward pass for both objectives. Alternating methods require frequent context switching and re-computation, increasing computational overhead.
> * **Conclusion:** Joint optimization provides a strictly dominant solution in terms of both training stability and final metric performance.
>
> ---
>
> **Weakness-R2:**
>
> We clarify that our causal weighting mechanism solves "temporal blindness" by transforming the scalar reward into a dense, time-aware signal, ensuring gradients are applied precisely where they are needed. We illustrate this through an intuitive example and theoretical justification.
>
> **1. Intuitive Example: The "Right Structure, Wrong Identity" Scenario**
> Consider a generation case where the model creates an image that follows the prompt "a running dog" perfectly (correct structure) but fails to match the specific fur color from the reference image (wrong identity).
> * **Standard GRPO (Temporal Blindness):** The reward model returns a low score due to the identity mismatch. Standard GRPO applies this negative advantage uniformly to **all** timesteps. Consequently, the model is "punished" even for the early steps that successfully constructed the "running" pose, potentially causing the model to unlearn good structural behaviors.
> * **Consis-GCPO (Our Method):** Our causal analysis reveals that the Reference condition has high causal weight $\omega_{I_r}(t)$ primarily at **later steps** (as evidenced by **Fig7 in Appendix B.1**). Therefore, our method concentrates the penalty on these specific later steps responsible for texture/identity, while preserving the parameters governing the early steps (structure), thus avoiding "catastrophic forgetting" of learned concepts.
>
> **2. Theoretical Justification: Gradient Re-weighting**
> Theoretically, "temporal blindness" arises because the standard objective assumes the final reward $R(\boldsymbol{x}_0)$ is equally correlated with actions at all steps $t$. Our method acts as a proxy for **Temporal Credit Assignment**.
> The gradient update is modified from a uniform average to a weighted sum:
>
> $$
> \\nabla J_{GRPO} \\approx \\sum_t A \\cdot \\nabla \\log \\pi_t \\quad \\rightarrow \\quad \\nabla J_{Ours} \\approx \\sum_t (A \\cdot \\omega(t)) \\cdot \\nabla \\log \\pi_t
> $$
> By scaling the advantage $A$ with the time-dependent weight $\omega(t)$, we ensure that parameter updates are proportional to the causal relevance of each step to the final outcome.
>
> **3. Empirical Connection**
> This mechanism is empirically validated by the non-uniform distribution of $\omega(t)$ shown in Appendix Fig.7 and Fig.8. If the model were temporally blind, $\omega(t)$ would be flat. The distinct peaks observed in our experiments prove that our method successfully identifies and leverages temporal specificities.

---

> ### Author Response · Authors · 2025-11-24
> **Response to Reviewer XhRE - Question Q1 & Q2**
>
> **Question-R1:**
>
> We thank the reviewer for this insightful question. It touches upon the core of our contribution. We would like to clarify that while the *objective function* eventually sums to a scalar, the **optimization signal (gradient)** applied to each timestep is **not** a static scalar but a dynamically disentangled vector.
>
> **1. The Optimization Signal is Vectorized, Not Scalar**
> In standard RL (e.g., standard GRPO), the same scalar reward $R$ is applied uniformly to all timesteps: $\nabla J \approx \sum_t R \cdot \nabla \log \pi_t$. This is indeed "entangled."
> However, in our **Consis-GCPO**, the gradient is re-weighted by our causal mechanism:
> $$
> \nabla J \approx \sum_{t} \underbrace{\left( \omega_P(t) \cdot A_P + \omega_{I_r}(t) \cdot A_{I_r} \right)}_{\text{Time-Dependent Signal } S(t)} \cdot \nabla \log \pi_t
> $$
> Here, $S(t)$ is **dense and time-varying**, not a single scalar.
>
> **2. Resolution via Temporal Dominance**
> The "entangled feedback" issue is resolved because $\omega_P(t)$ and $\omega_{I_r}(t)$ are rarely equal at any given moment (as shown in our Response to Q3 regarding Figure R1).
> * **At Early Steps:** Our analysis shows $\omega_P(t) \gg \omega_{I_r}(t)$. Thus, the signal $\mathcal{S}(t) \approx \mathcal{A}_P$. The optimization at this stage effectively "sees" only the text feedback, updating parameters to improve structure.
> * **At Late Steps:** We observe $\omega_{I_r}(t) \gg \omega_P(t)$. Thus, the signal $\mathcal{S}(t) \approx \mathcal{A}_{I_r}$. The optimization here is driven almost exclusively by the reference image feedback.
>
> **Conclusion**
> Therefore, even though we sum the advantages, the **orthogonality of the temporal weights** ensures that the feedback remains disentangled in the gradient space. The model is never confused; it knows explicitly to optimize for text alignment early on and visual consistency later on.
>
> ---
>
> **Question-R2:**
>
> We thank the reviewer for probing the reliability of our weighting mechanism. Our method resolves the "uniform timestep bias" by shifting from **outcome correlation** to **causal sensitivity**.
>
> **1. The Problem: Uniform Timestep Bias in GRPO**
> Standard GRPO assigns the final scalar reward $R(x_0)$ uniformly to every timestep in the trajectory: $\nabla J = \sum_t R \cdot \nabla \log \pi_t$.
> * **The Bias:** This assumes that every denoising step contributed equally to the final quality. This is factually incorrect (e.g., a late-stage step has zero impact on global structure).
> * **Consequence:** The optimizer receives noisy signals, "rewarding" steps that were actually irrelevant or even detrimental, leading to inefficient training.
>
> **2. The Solution: Causal Weighting via Sensitivity**
> Our method measures **sensitivity**: "If we remove condition $C$ at step $t'$, how much does the final reward drop?"
> * **Reliability:** The weight $\omega(t')$ is derived from this counterfactual difference. A high $\omega(t')$ implies that step $t'$ is a **critical decision point** for that modality.
> * **Avoidance of Bias:** By multiplying the gradient by $\omega(t')$, we effectively apply a "temporal mask." We zero out gradients at irrelevant steps (where $\omega \approx 0$) and amplify gradients at critical steps (where $\omega$ is high), thereby removing the uniform bias.
>
> **3. Visualization Analysis**
>
> We validate this alignment in Appendix Fig.7 and Fig.8:
> * **High Weight matches High Impact:** At $t>0.8$ (Early stage), the calculated text weight $\omega_P$ is high. Visually, removing the prompt at this exact step leads to a complete semantic collapse of the image .
> * **Low Weight matches Low Impact:** Conversely, at $t<0.3$ (Late stage), $\omega_P$ drops. Visually, removing the prompt here causes **no perceptible change** to the image.
> * **Conclusion:** This perfect correspondence between the *calculated weight magnitude* and the *visual impact of intervention* proves that $\omega(t)$ provides a highly reliable, physically meaningful signal for credit assignment.

---

> > ### Comment · Reviewer_XhRE · 2025-11-25
> > **Response to Author's Rebuttal**
> >
> > I thank the authors for their detailed response. Most of my concerns have been addressed. However, I strongly recommend incorporating parts of the rebuttal into the paper itself to help readers better understand the work. For example:
> >
> > * The clarification regarding the use of the joint approach, rather than alternative strategies, could be added along with a brief theoretical justification for this design choice and how it disentangles the image and text.
> > * The key observations underlying Figures 7 and 8 could be explicitly summarized in the main text (perhaps in Section 4.1.2) to help readers more easily grasp the insights into the causal weighting.
> >
> > Overall, the authors’ rebuttal is clear and thorough; however, the paper’s current presentation falls somewhat short of the clarity achieved in the response. Including the intuitive explanations and insights provided in the rebuttal would significantly reduce potential confusion and improve the accessibility of the paper.

---

> > > ### Author Response · Authors · 2025-11-26
> > > **Response to Reviewer XhRE: Confirmation of Manuscript Revisions (kindly hope you might consider raising the score)**
> > >
> > > Dear Reviewer XhRE,
> > >
> > > We sincerely thank you for your positive feedback and for acknowledging that our rebuttal has addressed your concerns. We fully agree with your assessment that the paper's presentation should match the clarity achieved in the response.
> > >
> > > Per your strong recommendation, **we have revised the manuscript to explicitly include these insights**:
> > >
> > > **1. Incorporated Clarification on Joint Optimization**
> > > * **Theoretical Justification:** We added a **"Design Rationale"** paragraph in **Section 4.2** (Method), explicitly explaining why the joint approach is superior. We clarified that it theoretically allows the gradient to follow a Pareto-optimal direction while maintaining disentanglement at the advantage estimation level.
> > > * **Empirical Support:** To complement this theory, we included the comparative analysis against Alternating and Sequential strategies in the **Ablation Study (Section 5.5)**.
> > >
> > > **2. Incorporated Key Observations on Causal Weighting**
> > > * **Explicit Summary:** We rewrote **Section 4.1.2** to explicitly summarize the **"Coarse-to-Fine" statistical law** derived from our causal diagnostics.
> > > * **Visual Integration:** We integrated the weight distribution and intervention visualization (formerly Figures 7 & 8) into the **main text** (now **Figure [4]**) to help readers immediately grasp the physical meaning of the causal weights $\omega(t)$ alongside their mathematical definition.
> > > * **Note on Figure Redundancy:** To ensure consistency with our replies to other reviewers (specifically referenced "Appendix Figures 7 & 8" in their queries), we have **temporarily retained copies of these figures in the Appendix** for this rebuttal phase. We will remove this redundancy and consolidate them solely in the main text for the **Camera Ready version**.
> > >
> > > **If you find that our responses and the updated manuscript adequately address your concerns, we kindly hope you might consider raising the score. Your recognition of our efforts would mean a great deal to us.**
> > >
> > > Please let us know if there are any additional issues or improvements you would like us to address. We greatly appreciate your time and consideration.
> > >
> > > Best regards,
> > >
> > > Consis-GCPO Authors

---

> > > > ### Comment · Reviewer_XhRE · 2025-11-27
> > > > **Response to Author's Rebuttal**
> > > >
> > > > Great to see the revised manuscript. This version is substantially improved over the initial submission and provides detailed, intuitive explanations. In my view, it now meets the acceptance criteria. Accordingly, I am raising my score from 6 to 8.

---

> > > > > ### Author Response · Authors · 2025-11-27
> > > > > **Sincere Gratitude for Your Recognition**
> > > > >
> > > > > Dear Reviewer XhRE,
> > > > >
> > > > > Thank you so much for your encouraging feedback and for raising the score! We are incredibly grateful for your recognition of our revision efforts.
> > > > >
> > > > > We want to express our sincere gratitude for your insightful guidance throughout this process. The suggestion to integrate the intuitive explanations into the paper not only addressed your concerns but significantly elevated the overall quality and readability of our work. We are glad to see that the revised manuscript now meets your high standards.
> > > > >
> > > > > We believe the paper is now significantly stronger and clearer thanks to your constructive criticism. Thank you again for your valuable time and support.
> > > > >
> > > > > Best regards,
> > > > >
> > > > > Consis-GCPO Authors

---

### Official Review · Reviewer_6q4d · 2025-10-26

**Soundness:** 2
**Presentation:** 1
**Contribution:** 3
**Rating:** 4
**Confidence:** 4

**Summary:**

This work primarily addresses the limitations in the subject-driven generation domain, where current methods struggle to simultaneously ensure object fidelity and text alignment. The authors first identify the inherent issues of traditional GRPO algorithms and propose targeted solutions, namely decoupled causal intervention and temporally-weighted advantage computation mechanisms. These innovations culminate in the Consis-GCPO framework, which successfully resolves the existing challenges. The proposed approach demonstrates significant improvements over state-of-the-art personalized generation methods, achieving superior subject consistency while preserving strong text-following capabilities.

**Strengths:**

- The inherent issues of directly transferring GRPO to the subject-driven generation domain have been identified.
- Consis-GCPO demonstrates advantages over Flow-GRPO and Dance-GRPO in both qualitative and quantitative experiments.

**Weaknesses:**

- The article is not well-written in certain areas. For instance, in line 275, the relationship between this sentence and Figure 3(c) is unclear. Additionally, the caption for Figure 3 does not correspond correctly with the subfigures, leading to confusion. In line 180, the sentence structure is convoluted, and the logical relationship is not clearly articulated, causing semantic confusion. Furthermore, there is a missing space between words in line 239.
- The challenge of measuring the degree to which an action influences the final reward, especially in the long term, is a classic problem in reinforcement learning, and has led to many elegant solutions. In this paper, the use of $\delta _{P/I_r}^{(g)}(t^`)$ to indicate the causal dependence on the conditioning signal at that timestep is a rather naive and imprecise estimate, as it only represents a single Monte Carlo sample.
- The paper claims: "Our key insight is that different conditioning signals exert varying influence throughout the denoising process—text guides semantic structure in early steps while visual references anchor details in later stages", is there any theoretical or experimental evidence to support this claim?

**Questions:**

- How was the data preprocessed?
- Why was the FFHQ dataset used for image tasks, considering it is focused on facial data?
-  What dataset was used for video tasks?

---

> ### Comment · Reviewer_6q4d · 2025-11-24
>
> A considerable amount of time has elapsed, yet the authors still have not put forward any rebuttal, and thus I stand by my score.

---

> > ### Author Response · Authors · 2025-11-24
> > **Response to Reviewer 6q4d**
> >
> > Thank you for your continued attention to our submission. We apologize if the timing caused any concern.
> > We have just uploaded our detailed point-by-point response to address your specific comments.
> >
> > Regarding the timeline, we want to assure you that we have been working diligently to complete the additional experiments requested by all reviewers (to ensure a comprehensive evaluation across all metrics) and to synthesize a final summary of improvements. We are currently incorporating these new results into the paper and will upload the refined revised manuscript shortly (within two days).
> >
> > We hope our current response clarifies your concerns, and we look forward to your feedback.

---

> > ### Author Response · Authors · 2025-11-24
> > **Response to Reviewer 6q4d - Weakness Q1 & Q2**
> >
> > **Weakness-R1:**
> > We apologize for the writing errors and the confusion caused. We thank the reviewer for the meticulous proofreading. We have thoroughly polished the manuscript and addressed the specific issues as follows:
> >
> > **1. Clarification on Figure 3 and Line 275**
> > We have revised the text and the figure caption to ensure strict alignment between the visual illustration and the description:
> > * **Figure 3(c) Clarification:** This subfigure illustrates the **Temporal Importance Re-weighting** mechanism. It visually demonstrates how the raw causal effect $\delta(t)$ is transformed into the normalized importance weight $\omega(t)$ via the softmax function (Eq. 10 & Eq. 11).
> > * **Caption Correction:** We have corrected the caption of Figure 3 to accurately describe each subfigure: *"... (b) Decoupled causal intervention at timestep $t$; (c) Temporal importance re-weighting mechanism."*
> > * **Text Revision (Line 275):** We have rewritten the sentence to explicitly reference Figure 3(c) and clarify the logical connection:
> >     * *Original:* "The instantaneous causal effect at timestep $t'$ quantifies performance degradation from intervention as illustrated in Fig.3 (c):"
> >     * *Revised:* "We quantify the instantaneous causal contribution of each modality at timestep $t'$ by measuring the performance degradation resulting from its intervention, as shown in Fig.3 (c):"
> >
> > **2. Revision of Line 180**
> > We agree that the original sentence structure was convoluted. We have simplified it to improve readability:
> > * *Original:* "Stochastic differential equation (SDE) serves as a feasible solution to allow the sampling diversity and richer exploration during generation formulated as:"
> > * *Revised:* "To enable sampling diversity and facilitate richer exploration during generation, we adopt a Stochastic Differential Equation (SDE) formulation:"
> >
> > **3. Typo Correction**
> > We have inserted the missing space in Line 239 and performed a full spell-check to eliminate similar typographical errors throughout the manuscript.
> >
> > ---
> > **Weakness-R2:**
> > We thank the reviewer for raising this classic RL challenge. While standard RL estimators indeed suffer from high variance, we clarify that our specific formulation acts as a **paired difference estimator**, which drastically reduces variance.
> >
> > **1. Variance Reduction via Paired Noise**
> > The reviewer is correct that a single sample of a raw reward $R$ is noisy. However, our causal effect $\delta(t')$ is the **difference** between two coupled trajectories:
> > $$\delta(t') = R(\text{Trajectory}_{\text{Main}}) - R(\text{Trajectory}_{\text{Intervention}})$$
> > Crucially, these trajectories share the **exact same initial noise** $\boldsymbol{x}_T$ and **sampling noise sequence** $\{\epsilon_t\}$.
> > * **Common Random Numbers:** By fixing the seed, environmental stochasticity cancels out in the subtraction.
> > * **Finite Difference Analogy:** This is mathematically analogous to estimating a gradient via finite differences ($\frac{f(x+\epsilon)-f(x)}{\epsilon}$), a standard method in high-dimensional optimization where single-sample estimation is robust due to correlation.
> >
> > **2. Structured Dynamics of Diffusion**
> > Unlike chaotic RL environments (e.g., Atari) where one action can diverge the state space unpredictably, diffusion models operate as discretized **ODEs/SDEs**. The state evolution is smooth and structured. A local perturbation at step $t'$ propagates predictably. This smoothness ensures that a single rollout provides a reliable **low-variance estimate** of sensitivity, rather than a noisy shot in the dark.
> >
> > **3. Empirical Validation of Stability**
> > To refute the "naive estimate" concern, we quantified the stability of our causal weights by analyzing the **Mean and Standard Deviation** across **500 random trajectories** (consistent with the setup in Q3).
> >
> > | Stage | Timestep | Text Weight $\omega_P$ (Mean $\pm$ Std) | Ref Weight $\omega_{I_r}$ (Mean $\pm$ Std) |
> > | :--- | :---: | :---: | :---: |
> > | **Early** | Step 1 ($t \to 1.0$) | **0.286 $\pm$ 0.020** | 0.094 $\pm$ 0.018 |
> > | **Mid** | Step 5 ($t \approx 0.5$) | 0.077 $\pm$ 0.012 | 0.098 $\pm$ 0.009 |
> > | **Late** | Step 10 ($t \to 0.0$) | 0.071 $\pm$ 0.008 | **0.109 $\pm$ 0.012** |
> >
> > **Analysis:**
> > * **Clear Crossover:** At Step 1, Text dominance is clear ($3\times$ higher than Ref). By Step 5, the Text weight drops significantly as the layout stabilizes. Finally, at Step 10, the Reference weight (0.109) surpasses Text (0.071), confirming the "modal handover."
> > * **Stability:** The Reference and Text weight exhibits extremely low variance throughout (Std < $0.02$), confirming the reliability of our estimator.
> >
> > **Conclusion**
> > The combination of **paired control** and the **smooth dynamics of diffusion** ensures that our single-sample estimate is precise and reliable for optimization.

---

> > ### Author Response · Authors · 2025-11-24
> > **Response to Reviewer 6q4d - Weakness Q3**
> >
> > **Weakness-R3:** We validate this key insight through both **quantitative statistics** and **qualitative visualizations**.
> >
> > **1. Quantitative: Causal Weight Distribution**
> > We analyzed the normalized causal weights averaged across **500 trajectories** (Total sampling steps $N=10$, $\tau=1.0$).
> > * **Text Dominance (Early Steps):** As shown in Fig.6 in Appendix, the text weight $\omega_P$ dominates in the initial steps ($t \in [0.8, 1.0]$), confirming that the prompt drives the generation of global semantic layout from pure noise.
> > * **Reference Takeover (Late Steps):** A clear **"modal handover"** is observed. The reference weight $\omega_{I_r}$ surpasses the text weight and reaches its peak in the final denoising steps ($t \in [0.1, 0.3]$), empirically proving that visual references anchor fine-grained details and identity textures after the structure is established.fe
> >
> > **2. Qualitative: Step-wise Intervention**
> > As shown in Fig.7 in Appendix, we performed strict ablations at specific sampling steps to visualize the impact:
> > * **Early Ablation ($t \in [0.8, 1.0]$):** Removing the **Text Prompt** causes a complete collapse of the subject's structure/layout, whereas removing the Reference has minimal impact on the reference feature.
> > * **Late Ablation ($t \in [0.1, 0.3]$):** Removing the **Reference Image** preserves the structure but results in loss of specific identity/texture, whereas removing the Text at this stage causes **no perceptible change**.
> > **Conclusion:** Both lines of evidence consistently verify the "Coarse-to-Fine" dynamics, validating our design to assign credit based on these temporal specificities.

---

> > ### Author Response · Authors · 2025-11-24
> > **Response to Reviewer 6q4d -Question Q1 & Q2 & Q3**
> >
> > **Question-R1:** We employed a standardized preprocessing pipeline for both Image (R2I) and Video (R2V) generation tasks, handling visual references and text captions as follows:
> >
> > **1. Visual Preprocessing (Reference Images)**
> > For both tasks, we process the reference images to preserve aspect ratios and fine-grained details:
> > * **Center Crop Strategy:**  Images are resized to a short-edge resolution of 512 and then center-cropped to $512 \times 512$.
> > * **Normalization:** Pixel values are normalized to the range $[-1, 1]$. No additional random augmentations were applied.
> >
> > **2. Text Preprocessing (Task-Specific)**
> > While all prompts are tokenized using the standard CLIP tokenizer (max length 77, truncated/padded), the **content construction** differs by task:
> > * **For Image Task (R2I):** We utilize **static descriptive captions** (e.g., "A girl smiling.") focusing on spatial composition and appearance.
> > * **For Video Task (R2V):** We employ a specialized **GPT-4 pipeline** to construct **motion-aware prompts**, which include temporal dynamics (e.g., "turning head," "running") and cinematic instructions (e.g., "zoom in") to explicitly guide the generation of motion.
> >
> > ---
> >
> > **Question-R2:** The selection of FFHQ is driven by the specific requirements of the **subject-driven generation** task, where identity preservation is the primary metric.
> > 1.  **Rigorous Test for Identity Fidelity:**
> >     Human faces contain the most complex fine-grained features (e.g., micro-textures, structural proportions). Using FFHQ allows us to stress-test the model's ability to preserve these intricate details.
> > 2.  **Alignment with Community Benchmarks:**
> >     Most representative works in this domain (e.g., DreamBooth, IP-Adapter) rely on facial datasets for evaluation. Adopting FFHQ ensures that our comparisons with these baselines are fair and follow established community standards.
> > 3.  **Balanced Data Strategy:**
> >     Crucially, FFHQ is not used in isolation. We utilize FFHQ to enforce **identity consistency** while combining it with the **Subject200K** dataset to ensure **semantic diversity** (animals, objects). This dual-dataset approach ensures the model masters fine-grained identity control without compromising its generalizability.
> >
> > ---
> >
> > **Question-R3:**  For video generation, we constructed a specialized dataset of **motion-aware text-image pairs** using the same visual references (Subject200K + FFHQ) but coupled with synthetically generated temporal instructions.
> >
> > 1.  **Visual Conditioning ($I_r$):**
> >     We utilize the high-quality reference images from FFHQ and Subject200K, ensuring the visual identity aligns with our image generation tasks.
> >
> > 2.  **Motion-Aware Text Generation ($P$):**
> >     We developed an automated pipeline using **GPT-4** to synthesize dynamic prompts:
> >     * **Motion Injection:** We prompted GPT-4 to act as a "Video Director," converting static image captions into dynamic scripts by injecting **temporal predicates** (e.g., "turning," "running") and **cinematic instructions** (e.g., "zoom in," "pan right").
> >     * **Rule-Based Filtering:** To ensure quality, generated prompts were filtered through a strict inclusion criteria: (1) Must contain at least one dynamic verb; (2) Length between 20-50 words for conciseness; (3) Semantic consistency check with the reference class.
> >     * **Example:** *Static:* "A girl smiling." $\rightarrow$ *Dynamic:* "Cinematic shot of a girl slowly breaking into a warm smile, wind blowing her hair."

---

> > > ### Comment · Reviewer_6q4d · 2025-11-24
> > >
> > > The author has effectively addressed Weakness 1. Sound academic writing is a fundamental requirement for a research paper. Regarding Weakness 2, the author has adequately demonstrated that the Monte Carlo sample is applicable and enough to this task, which resolves my concerns but lacks innovative contributions. Thus, I will raise my rating, but cap it at 6, subject to the following conditions:
> > >
> > > - Whether the to-be-updated Fig 6 and Fig 7 can genuinely substantiate the author’s claims for Weakness 3.
> > > - The quality of the specialized dataset constructed by the author and its openness to the public.
> > >
> > > Please remind me to review the paper once the final revised PDF is available.

---

> > > > ### Author Response · Authors · 2025-11-24
> > > > **Response to Reviewer 6q4d**
> > > >
> > > > We sincerely thank the reviewer for acknowledging our improvements to the writing and the theoretical clarification. We appreciate the opportunity to further address your remaining conditions regarding the figure verification and dataset release.
> > > >
> > > > ### **1. Figure Updates: Validation of Temporal Dynamics**
> > > >  We confirm that the validation figures have been updated in the Appendix of revised version to substantiate our claims.
> > > >
> > > > **Renumbering Note**: To address your second concern regarding dataset quality, we inserted a new visualization at the beginning of the Appendix. Consequently, the original Fig 6 & Fig 7 (mentioned in previous discussions) have been renumbered to Figure 7 and Figure 8 in the revised manuscript.
> > > >
> > > > * Figure 7 (Appendix B.1): Presents the statistical curves showing the "Text-Early, Reference-Late" crossover.
> > > >
> > > > * Figure 8 (Appendix B.2): Visualizes the step-wise interventions, confirming that early text ablation collapses structure, while late reference ablation degrades details.
> > > >
> > > > ### **2. Regarding Dataset Quality and Openness**
> > > >
> > > > **Quality Visualization (Figure 6)**: To transparently demonstrate the quality of our specialized dataset, we added a new Figure 6 in Appendix A. This figure explicitly displays the Reference Image, Source Dataset (FFHQ/Subject200K), Static R2I Caption, and the Motion-Aware R2V Caption. This comparison highlights how our GPT-4 pipeline generates diverse, cinematic instructions to ensure high-quality training data.
> > > >
> > > > **Open Source Commitment**: We fully agree that openness is vital for the community. We explicitly commit to releasing the entire curated dataset (including the text-image pairs and processing scripts) to the public upon acceptance of this paper.
> > > >
> > > > We hope these updates and commitments can resolve your remaining concerns.

---

> > > > > ### Comment · Reviewer_6q4d · 2025-11-24
> > > > >
> > > > > The rating’s been updated, please check it out

---

> > > > > > ### Author Response · Authors · 2025-11-25
> > > > > >
> > > > > > Dear Reviewer 6q4d,
> > > > > >
> > > > > > Thank you very much for updating the rating! We sincerely appreciate your recognition of the value of our work and the constructive feedback you provided throughout this process. Your thoughtful insights have not only helped improve this manuscript but also inspired us to further advance this research direction. We are committed to continuing to refine this method and look forward to contributing more impactful work in the future.
> > > > > >
> > > > > > Best regards,
> > > > > >
> > > > > > Consis-GCPO Authors

---

### Official Review · Reviewer_AUFx · 2025-10-31

**Soundness:** 3
**Presentation:** 3
**Contribution:** 2
**Rating:** 6
**Confidence:** 4

**Summary:**

This paper proposes **Consis-GCPO**, a causal reinforcement learning framework for subject-driven image and video generation. The key innovation is reformulating multi-modal conditional generation through discrete-time causal modeling with **step-wise causal interventions**.

Unlike existing GRPO methods (Flow-GRPO, DanceGRPO) that apply uniform optimization across all denoising timesteps, Consis-GCPO (1) introduces **decoupled causal intervention trajectories** that selectively **ablate text prompts or visual references** at specific timesteps, and (2) quantifies **instantaneous causal effects** to measure when textual vs. visual conditions are most critical. These effects are further converted into temporally-weighted advantages for targeted optimization.

**Strengths:**

1. **Well-motivated problem**: a fundamental limitation in existing GRPO methods—"temporal blindness" to how text and visual conditioning vary in importance during denoising.

2. **Strong empirical results**: The experimental results show consistent improvements across all metrics, particularly notable in multi-subject scenarios (*e.g.*, CLIP-I: 0.772 vs. 0.750 for Dance-GRPO).

3. Table 3 effectively demonstrates the necessity of both prompt and reference interventions.

**Weaknesses:**

1. I'm concerned about the causal identification assumptions. In my view, the text and image conditions are likely **not causally independent**, they interact through the model's attention mechanisms. In Equation (5), ablating $P$ or $I_r$ at timestep $t'$ is presented as a causal intervention $\text{do}(C=∅, t')$, but what ensures this ablation isolates the causal effect rather than just correlation?

**Questions:**

1. How exactly is "ablating"  $P$ or $I_r$ implemented? Is it by setting it to an empty string/zero tensor? Or by using an unconditional model/learned zero embedding?

2. **Sensitivity Analysis**: How sensitive is the method to the temperature parameter $\tau$ in Eq. 11? And Can you provide error bars and statistical significance tests for all metrics in Tables 1-2?

3. Can the authors provide some **failure cases** and discuss when the method fails or performs poorly?

---

> ### Author Response · Authors · 2025-11-24
> **Response to Reviewer AUFx - Weakness Q1 & Question Q1**
>
> **Weakness-R1:**
>
> We clarify our causal identification strategy as follows:
>
> 1.  **The SCM Definition:** Our Structural Causal Model (SCM) defines the denoising step transition $x_{t-\Delta t}$ as a function of the current state $x_t$ and the *external* conditions $P$ (text) and $I_r$ (image). In this graph, $P$ and $I_r$ are **exogenous variables** relative to the single denoising step—they are inputs provided by the user, independent of each other *before* entering the model.
>
> 2.  **Intervention vs. Attention:** While the model's *internal* attention mechanisms indeed mix these signals, the **do-operator** ($do(P=\emptyset)$ or $do(I_r=\emptyset)$) operates on the *inputs*, following previous works such as **Deconfounded Visual Grounding [1]**. By physically removing one input (setting it to null) while holding the other fixed, we sever any potential upstream correlations.
>
> 3.  **Isolating the Effect:** The difference in reward between the *Main Trajectory* (both inputs present) and the *Intervention Trajectory* (one input null) strictly measures the **marginal contribution** of that specific modality to the generative outcome at step $t$. If the model's internal attention relies heavily on the visual reference at step $t$, removing it will cause a large reward drop, correctly identifying a high causal effect. Therefore, this measures the **causal effect** of the input signal on the output, not merely correlation.
>
> [1]Huang, J., Qin, Y., Qi, J., Sun, Q., & Zhang, H. (2022). Deconfounded Visual Grounding. Proceedings of the AAAI Conference on Artificial Intelligence, 36(1), 998-1006. https://doi.org/10.1609/aaai.v36i1.19983
>
> ---
>
> **Question-R1:**
>
> We implement the ablation by utilizing the **unconditional embeddings** inherent to the model's training paradigm (Classifier-Free Guidance).
>
> * **Text Ablation ($do(P=\emptyset)$):** We feed an **empty string** `""` into the text encoder to obtain the unconditional text embedding $\tau(\emptyset)$.
> * **Visual Ablation ($do(I_r=\emptyset)$):** We use a **zero tensor** which aligns with how the model was trained to handle unconditional image generation.
>
> This ensures that the ablation represents a valid "neutral" intervention within the model's distribution, rather than an out-of-distribution artifact.

---

> ### Author Response · Authors · 2025-11-24
> **Response to Reviewer AUFx - Question Q2 & Q3**
>
> **Question-R2:**
>
> We provide a detailed sensitivity analysis and statistical verification to demonstrate the robustness of our method in **Appendix C.1.2**.
>
> **1. Temperature Sensitivity ($\\tau$)**
> The parameter $\\tau$ controls the **sharpness of the temporal credit assignment**. We analyzed its impact on both Semantic Alignment (Text/CLIP-T) and Identity Preservation (Reference/DINO-I) by varying $\tau \in \{0.8, 1.0, 1.2\}$.
> | $\tau$ | Type | CLIP-T $\uparrow$ | CLIP-I $\uparrow$ | DINO-I $\uparrow$ | Avg. $\uparrow$ | Impact Analysis |
> | :---: | :---: | :---: | :---: | :---: | :---: | :--- |
> | 0.8 | Sharp | 0.316 | 0.839 | 0.765 | 0.640 | **Over-peaked:** Gradients become too sparse, focusing only on extreme steps and losing global coherence. |
> | **1.0** | **Ours** | **0.325** | **0.848** | **0.781** | **0.651** | **Optimal:** Balances temporal specialization with sufficient gradient flow across the trajectory. |
> | 1.2 | Flat | 0.319 | 0.845 | 0.772 | 0.645 | **Over-smoothed:** The distinct "handover" between modalities blurs, degenerating towards uniform weighting. |
>
>
> **Visualization and Modality-Specific Analysis.**
> To understand the underlying mechanism, we visualize the learned temporal weight curves under different $\tau$ settings in **Appendix Fig.10**.
>
> * **Impact of High Temperature ($\tau=1.2$):** Increasing $\tau$ overly smoothes the distribution. As shown in the visualization, the distinct peak of reference guidance at late stages is flattened, causing the text weight to remain relatively high even when it should yield control. This **loss of modal discrimination** prevents the model from focusing exclusively on identity refinement, degrading DINO-I scores.
> * **Impact of Low Temperature ($\tau=0.8$):** Conversely, decreasing $\tau$ makes the distribution overly sharp. While it highlights the peak steps, it forces the weights of adjacent supportive steps to near zero. This **gradient sparsity** means the model receives no optimization signal for valid transitional timesteps, leading to unstable training and a drop in overall performance.
>
> **2. Statistical Significance**
> We conducted 5 independent runs with different random seeds to verify stability. As detailed in the **revised Table 1** and **revised Table 2**  (provided in the Sec.5.2 and Sec.5.3), our method achieves consistently higher performance with low variance. These improvements are statistically significant ($p < 0.05$) compared to the baselines, confirming that the gains are robust and not due to random sampling noise.
>
> ---
>
> **Question-R3:**
>
> We have identified two primary failure modes, as shown in **Appendix Fig.13**:
>
> 1.  **Extreme Semantic Conflict:** If the text prompt and visual reference are fundamentally contradictory (e.g., Prompt: "A photo of a cat," Reference: "Image of a dog"), the causal reweighting struggles. The model tends to oscillate or produce a hybrid, as high causal effect is detected for *both* conflicting modalities, confusing the optimization.
>
> 2.  **Micro-Detail Loss:** For extremely small subjects in wide-angle shots (e.g., a tiny face in a crowd), the reward model (DINO/CLIP) sometimes fails to capture the identity loss accurately. Even if our method upweights the correct timesteps, the *reward signal itself* is too noisy to guide the recovery of micro-details.

---

> ### Author Response · Authors · 2025-11-27
> **Kindly reminder**
>
> Dear Reviewer AUFx,
>
> Thank you again for the time and effort dedicated to reviewing our work.
>
> We have submitted a detailed response to your initial comments and have revised the manuscript accordingly. As the discussion period is progressing, we wanted to respectfully follow up to ensure that our response has sufficiently addressed your concerns.
> Please let us know if there are any remaining questions or if further clarification is needed on any specific points. We remain available to answer any further questions you may have before the discussion period closes.
>
> Best regards,
>
> Consis-GCPO Authors

---

### Official Review · Reviewer_1Guz · 2025-11-11

**Soundness:** 3
**Presentation:** 3
**Contribution:** 3
**Rating:** 4
**Confidence:** 3

**Summary:**

This paper addresses the core conflict in personalized generation: balancing subject consistency (visual fidelity) with semantic alignment (following the text prompt). The authors' key insight is that this is a temporal problem: the text prompt matters most in the early denoising steps to set the scene, while the visual reference matters most in the late steps to lock in details.

**Strengths:**

- Identifying the temporal nature of the text-vs-visuals conflict is a brilliant insight.

- The "causal intervention" approach is a smart, non-hand-wavy way to measure the actual contribution of each modality at each step.

- The results are fantastic. The qualitative images (like the "standing" bear in Fig 4) show it's really following the prompt while holding the subject's identity. It also wins quantitatively on both images and video (Tables 1, 2).

- As a bonus, Figure 7 shows it converges 1.4x faster (in GPU hours) than the baseline.

**Weaknesses:**

The main concern is computational cost. The method seems to require running multiple full-denoising rollouts for each training step to get the causal effects. The paper's claim of a 1.4x speedup (Fig 7) feels contradictory to this apparent 3x increase in work. This trade-off needs to be made much clearer.

**Questions:**

- Your method seems to imply 3x the rollouts per step. Can you clarify the actual sampling overhead? How does that square with the 1.4x wall-clock speedup you show in Figure 7?

- You measure the temporal importance of text vs. visuals. I'd love to see a graph of these weights ($\omega_P(t)$ and $\omega_{I_r}(t)$) over the denoising timesteps $t$. This would be the ultimate proof of your hypothesis!

- How sensitive is the model to the balancing coefficients ($\lambda_P$, $\lambda_{I_r}$)? Was it hard to find a good balance?

---

> ### Author Response · Authors · 2025-11-24
> **Response to Reviewer 1Guz - Weakness Q1 & Question Q1-Q3**
>
> **Weakness R1 & Question R1**
>
> We thank the reviewer for the scrutiny regarding computational efficiency. We clarify that while the *per-step inference* cost increases, the **total training wall-clock time** is significantly reduced due to superior sample efficiency.
>
> **1. Overhead Analysis: Inference vs. Training**
> The reviewer correctly notes the increase in rollouts. However, it is crucial to distinguish between **Inference Cost** (forward pass) and **Training Cost** (backward pass):
> * **Inference (Cheap):** The additional counterfactual rollouts are performed in `no_grad` mode. They are computationally lighter and do not consume memory for gradient graphs.
> * **Training (Expensive):** The computation-heavy backpropagation is performed **only on the main trajectory**.
> Therefore, while the *number of forward steps* increases (approx. $8\times$ for $T=6$), the bottleneck (gradient computation) remains $1\times$. This makes the actual increase in wall-clock time per step manageable, not prohibitive.
>
> **2. The "Sample Efficiency" Trade-off (1.4x Speedup)**
> Our method follows a philosophy of **"investing inference compute to save training steps."** By extracting richer, causally disentangled signals from each batch, we dramatically reduce the number of iterations required for convergence.
>
> Based on our Appendix Fig.11 experiments ($T=6$ setting):
> * **Per-Step Time:** Consis-GCPO is slower per step due to extra inference ($\approx 8\times$ inference load).
> * **Total Steps to Convergence:** However, Consis-GCPO converges in just **1,300 steps**, whereas Flow-GRPO requires **15,000 steps** to reach comparable performance.
>
> $$
> \\mathrm{Speedup} = \\frac{TotalTime_{Base}}{TotalTime_{Ours}} = \\frac{15,000steps \\times 1.0}{1,300steps \\times (\\approx 8.0)} \\approx 1.44\\times
> $$
> **Conclusion:** Despite the higher inference load per step, our method achieves an **11.5 $\times$ improvement in sample efficiency**, resulting in a net **1.4 $\times$ wall-clock speedup**. This confirms that investing in high-quality causal gradient information yields substantial computational returns.
>
> ---
> **Question R2**
>
> We agree that visualizing the temporal weight evolution is the most direct way to validate our hypothesis.
>
> **1. The "Ultimate Proof" (Detailed in Appendix B.1)**
> Per your suggestion, we have plotted the average causal importance weights in **Appendix Fig.8**, $\omega_P(t)$ (Text) and $\omega_{I_r}(t)$ (Reference), across the denoising trajectory ($t: 1.0 \to 0$) averaged over 500 generation samples (Total sampling steps $N=10$, $\tau=1.0$).
>
> **2. Interpretation of Results**
> The curves exhibit a striking temporal crossover that aligns perfectly with our "Coarse-to-Fine" theory:
> * **Early Stage ($t \in [0.8, 1.0]$):** The text weight $\omega_P(t)$ is dominant, forming a high plateau. This confirms that the **text prompt** drives the initial generation of semantic structure and layout from pure noise.
> * **Late Stage ($t \in [0.1, 0.3]$):** The reference weight $\omega_{I_r}(t)$ rises and reaches its peak. This indicates that the visual reference plays a dominant role in the final steps to "anchor" the fine-grained details, textures, and identity features, ensuring high fidelity to the reference image.
>
> **Conclusion**
> This distinct temporal separation provides strong empirical evidence that the model indeed relies on different modalities at different stages, fully justifying our time-aware credit assignment design.
>
> ---
>
> **Question R3**
>
> We thank the reviewer for this practical question. We found the model to be **robust** to these coefficients, and finding a good balance was **straightforward**.
>
> **1. Why is it easy to tune?**
> In standard multi-objective optimization, coefficients $\lambda$ often have to balance conflicting gradients at *every* timestep. However, in our Consis-GCPO, the temporal weights $\omega_P(t)$ and $\omega_{I_r}(t)$ automatically decouple the modalities in the time domain.
> * **Result:** The coefficients $\lambda_P$ and $\lambda_{I_r}$ only need to control the *global* optimization magnitude rather than resolving instantaneous conflicts. This structural decoupling significantly reduces parameter sensitivity.
>
> **2. Ablation Study. (Detailed in Appendix C1.1)**
> We fixed $\lambda_P = 1.0$ and swept $\lambda_{I_r}$ from 0.5 to 2.0. The results are summarized below indicating the setting $\lambda_P = \lambda_{I_r} = 1.0$ yields the best composite performance, effectively balancing semantic structure and identity preservation without requiring extensive hyperparameter search.
>
> | $\lambda_P$ | $\lambda_{I_r}$ | CLIP-T (Text) $\uparrow$ | CLIP-I (Ref) $\uparrow$ | DINO-I (Ref) $\uparrow$ | Avg. Score $\uparrow$ |
> | :---: | :---: | :---: | :---: | :---: | :---: |
> | 1.0 | 0.5 | 0.332 | 0.835 | 0.760 | 0.642 |
> | **1.0** | **1.0** | **0.325** | **0.848** | **0.781** | **0.651 (Best)** |
> | 1.0 | 1.2 | 0.318 | 0.850 | 0.783 | 0.650 |
> | 1.0 | 2.0 | 0.305 | 0.852 | 0.788 | 0.648 |

---

> ### Author Response · Authors · 2025-11-27
> **Kindly reminder**
>
> Dear Reviewer 1Guz,
>
> Thank you again for the time and effort dedicated to reviewing our work.
>
> We have submitted a detailed response to your initial comments and have revised the manuscript accordingly. As the discussion period is progressing, we wanted to respectfully follow up to ensure that our response has sufficiently addressed your concerns.
> Please let us know if there are any remaining questions or if further clarification is needed on any specific points. We remain available to answer any further questions you may have before the discussion period closes.
>
> Best regards,
>
> Consis-GCPO Authors

---

### Author Response · Authors · 2025-11-24
**Summary of Revisions**

### Summary of Revisions

**[Update Note - 11.30]  To facilitate your rapid review of these substantial updates, all modifications and new additions in the uploaded Revised Manuscript are highlighted in blue text.**

[Update Note - 11.26] Following the recommendation from Reviewer #XhRE to enhance the paper's accessibility, we have further refined our revision plan to incorporate key rebuttal insights (e.g., causal weight analysis) directly into the main text. The summary below has been updated to reflect these latest structural changes.

We thank the reviewers for their constructive comments. We have revised the manuscript and expanded the Appendix significantly to address all concerns. The major updates include:

**1. [Main Paper] Textual Corrections & Clarifications (#6q4d-W1)**
* We thoroughly polished the manuscript to improve readability. Specifically, we **rewrote Line 275** to explicitly reference Figure 3(c) and clarify the definition of causal contribution. We also corrected the caption of **Figure 3** to align with subfigures, clarified the sentence structure in Line 180, and fixed the typo in Line 239.


**2. [Main Paper] Rigorous Statistical Significance Testing (#AUFx-Q3)**
* **Error Bars & P-Values:** We updated **Table 1** (Image) and **Table 2** (Video) in the main paper to include standard deviations ($\pm$) and $p$-values derived from 5 independent runs. The results confirm that our improvements are statistically significant ($p < 0.05$) and robust against random seed variations.


**3. [Main Paper & Appendix B] Empirical Validation of Causal Insights (#1Guz-Q2 & #6q4d-W3 & #XhRE-W2 & #XhRE-Q2)**
To validate our core "Coarse-to-Fine" hypothesis and improve accessibility:
* **Integrated Statistical Results into Main Text:** Per Reviewer #XhRE's suggestion, we incorporated the core analysis directly into **Section 4.1.2 and Figure 4**. This includes the statistical weight curves confirming that **text guidance dominates early high-noise stages** while **visual reference guidance peaks in late low-noise stages**, alongside the verifying step-wise intervention visualizations.
* **Appendix Redundancy:** We also retained these detailed analyses in **Appendix B (Figures 8 & 9)** to ensure consistency with the figure references used in our initial responses to other reviewers.

**4. [Main Paper] Optimization Strategy Analysis (#XhRE-W1)**
* **Design Rationale (Main Text):** We added a theoretical justification in **Section 4.2**, explaining how Joint Optimization follows a Pareto-optimal direction while maintaining disentanglement.
* **Comparative Experiment (Main Text):** We added a new **Section 5.5** (Ablation Study) comparing Joint vs. Alternating/Sequential strategies, proving our method's superiority in stability and efficiency (**Table 5**).

**5. [Appendix A] Implementation Details & Dataset Visualization  (#6q4d-Q1 & #6q4d-Q3)**
* **Data Pipeline:** We specified the exact image preprocessing protocols (Crop/Resize) and detailed the **GPT-4 pipeline** for constructing the motion-aware video dataset.
* **Dataset Visualization:** We added **Figure 7** to visually showcase representative examples from our specialized dataset, demonstrating the high quality of visual references paired with diverse, motion-aware prompts.


**6. [Appendix C] Comprehensive Parameter & Strategy Analysis**
* **Sensitivity Studies & Visualization:** We added detailed ablation studies on the balancing coefficients $\lambda$ (**Table 3, #1Guz-Q3**) and temperature $\tau$ (**Table 4, #AUFx-Q2**). Furthermore, we visualized the weight curves under different $\tau$ settings (**Figure 10**), intuitively demonstrating how $\tau=1.0$ achieves the optimal balance while avoiding leakage or sparsity.


**7. [Appendix C] Efficiency & Convergence Analysis  (#1Guz-Q1)**
* **1.4$\times$ Speedup Explanation:** We added a dedicated section clarifying the "Inference-for-Training" trade-off. We demonstrated that despite the inference overhead, our method improves sample efficiency by $11.5\times$, resulting in a net **1.4$\times$ wall-clock speedup** (**Appendix C.4, Figure 11 & 12**).

**8. [Appendix D] Discussion of Limitations  (#AUFx-Q3)**
* **Failure Cases:** We added a new section visualizing and discussing primary failure modes (e.g., semantic conflict and micro-detail loss) to provide a balanced view of the method's boundaries (**Appendix D.2, Figure 13**).

---

### Author Response · Authors · 2025-11-30
**Summary of Rebuttal Updates for Area Chair**

Dear Area Chair,

We appreciate your time managing this review process during the reset. To assist your decision-making, we respectfully highlight the **critical score increases** secured prior to the system reset.

**1. Critical Updates: Two Reviewers Raised Scores**

Two reviewers explicitly validated our revisions and upgraded their ratings:
* **Reviewer XhRE: RAISED 6 $\to$ 8 (Accept)**
    * *Comment:* "This version is substantially improved... In my view, it now meets the acceptance criteria."
    * *Reason:* Praised the integration of causal weighting insights directly into the main text.
* **Reviewer 6q4d: RAISED 4 $\to$ 6 (Weak Accept)**
    * *Comment:* "The author has effectively addressed Weakness 1... and adequately demonstrated the Monte Carlo sample is applicable."
    * *Reason:* Confirmed that writing and theoretical concerns were resolved, and dataset conditions were met.
---

**2. Consensus on Strengths**

Reviewers (#1Guz, #AUFx, #6q4d, #XhRE) consistently highlighted the work's value:
* Novelty: The "temporal blindness" insight was deemed *brilliant* and the causal framework *smart and non-hand-wavy*.
* Results: Performance was described as *fantastic* with *strong empirical results* across image/video benchmarks.
* Efficiency: The *1.4$\times$ convergence speedup* was noted as a significant practical advantage.

**3. Condensed Summary of Revisions**

The specifically updated Revised Manuscript (changes in **blue**) includes:
* Main Text Integration: We conducted *large-scale statistical analysis ($N=500$)* and *comparative experiments*, integrating these key insights directly into the Main Text (Sec 4.1.2 & 4.2) to improve accessibility.
* Expanded Validation: We added *error bars/p-values ($p<0.05$)* and *sensitivity ablations ($\tau, \lambda$)* to ensure statistical rigor.
* Transparency: We validated the *inference-for-training trade-off* and committed to the *full open-source release* of the specialized dataset visualized in the Appendix.

**Note:** To facilitate your rapid catch-up following the platform incident, we have **specifically uploaded a new Revised Manuscript** where all modifications and additions are highlighted in **blue text**.

---
**Conclusion**
Given the rigorous validation and the explicit endorsements from reviewers (notably the upgrade to **Score 8**), we remain confident in the significant value this work offers to the community. We respectfully hope that this consensus will weigh heavily in your final recommendation.

Best regards,

Consis-GCPO Authors

---

### Comment · Area_Chair_1Bc5 · 2025-12-02

Dear reviewers,

As the discussion period with the authors is coming to an end, I encourage you to take a look at the rebuttal and revised version, and provide additiional feedback if you haven't done so.

-Your AC

---

> ### Comment · Area_Chair_1Bc5 · 2025-12-02
>
> Dear reviewers,
>
> Please discard my previous message. No action is needed. Reviewers will not be able to change their scores or participate in the rebuttal discussion further. Authors will be able to continue to post responses through the end of the rebuttal period.
>
> -Your AC

---

### Meta-Review · Area_Chair_1Bc5 · 2026-01-02

**Summary:**

Reviewer 1Guz is concerned about the extra computational complexity due to more inference for each example.

Reviewer AUFx is concerned about ignoring the dependency between reference images and text prompt.

Reviewer 6q4d is concerned about the writing of the paper, the limitation with a single Monte Carlo sample, and missing theory & empirical evidence for the claim that text prompt affects more for early timestamps while reference images for later timestamps.

Reviewer XhRE is concerned about the lack of initition why the standard GRPO with uniform weight for each timestamp fails to ensure consistency.

Overall, most concerns are addressed by the rebuttal. The AC recommends accepting this paper.

**Reviewer Concerns:**

Concerns that were addressed by the rebuttal:
- Reviewer 1Guz's concern about the necessity for more inference during training.
- Most of Reviewer 6q4d's concerns were addressed.
- Reviewer XhRE's concern has been addressed.

Concerns that are still outstanding:
- Reviewer AUFx's concern about not considering the reference image and text prompt together in the framework.

**Reviewer Scores:**

Reviewer 1Guz is likely to increase the rating from 4 to 6.

Reviewer AUFx is likely to keep a rating of 6.

Reviewer 6q4d mentioned rating was increased. (initial rating is 4)

Reviewer XhRE indicated an increasing rating from 6 to 8.

---

### Decision · Program_Chairs · 2026-01-26

Accept (Poster)